# Learning Ground Displacement Signals Directly from InSAR-Wrapped Interferograms

**DOI:** 10.3390/s24082637

**Published:** 2024-04-20

**Authors:** Lama Moualla, Alessio Rucci, Giampiero Naletto, Nantheera Anantrasirichai

**Affiliations:** 1Center of Studies and Activities for Space (CISAS) “G. Colombo”, University of Padova, Via Venezia 15, 35131 Padova, Italy; 2TRE-ALTAMIRA s.r.l., 20143 Milan, Italy; alessio.rucci@tre-altamira.com; 3Department of Physics and Astronomy Galileo Galilei—DFA, Padova University, 35131 Padova, Italy; giampiero.naletto@unipd.it; 4Visual Information Laboratory, University of Bristol, Bristol BS1 5DD, UK; 5COMET, School of Computer Science, University of Bristol, Bristol BS8 1UB, UK

**Keywords:** Sentinel-1, ground displacements, wrapped interferograms, P-SBAS, Cosine K-NN

## Abstract

Monitoring ground displacements identifies potential geohazard risks early before they cause critical damage. Interferometric synthetic aperture radar (InSAR) is one of the techniques that can monitor these displacements with sub-millimeter accuracy. However, using the InSAR technique is challenging due to the need for high expertise, large data volumes, and other complexities. Accordingly, the development of an automated system to indicate ground displacements directly from the wrapped interferograms and coherence maps could be highly advantageous. Here, we compare different machine learning algorithms to evaluate the feasibility of achieving this objective. The inputs for the implemented machine learning models were pixels selected from the filtered-wrapped interferograms of Sentinel-1, using a coherence threshold. The outputs were the same pixels labeled as fast positive, positive, fast negative, negative, and undefined movements. These labels were assigned based on the velocity values of the measurement points located within the pixels. We used the Parallel Small Baseline Subset service of the European Space Agency’s GeoHazards Exploitation Platform to create the necessary interferograms, coherence, and deformation velocity maps. Subsequently, we applied a high-pass filter to the wrapped interferograms to separate the displacement signal from the atmospheric errors. We successfully identified the patterns associated with slow and fast movements by discerning the unique distributions within the matrices representing each movement class. The experiments included three case studies (from Italy, Portugal, and the United States), noted for their high sensitivity to landslides. We found that the Cosine K-nearest neighbor model achieved the best test accuracy. It is important to note that the test sets were not merely hidden parts of the training set within the same region but also included adjacent areas. We further improved the performance with pseudo-labeling, an approach aimed at evaluating the generalizability and robustness of the trained model beyond its immediate training environment. The lowest test accuracy achieved by the implemented algorithm was 80.1%. Furthermore, we used ArcGIS Pro 3.3 to compare the ground truth with the predictions to visualize the results better. The comparison aimed to explore indications of displacements affecting the main roads in the studied area.

## 1. Introduction

Infrastructure is one of the cornerstones of building societies. Hence, there is a critical need to monitor its performance through effective and reliable methodologies. Most of the infrastructure facilities are in direct contact with the soil. Thus, it is necessary to monitor the facilities and the areas surrounding their foundations, especially when the infrastructure is exposed to severe ground changes (such as landslides) as the risks to its stability and safety multiply.

Interferometric synthetic aperture radar (InSAR) is a technique that utilizes the coherence of the radar signals to measure the displacement and deformation of the Earth’s surface. This principle, as defined in [1], works by comparing two or more complex-valued SAR images of the same area taken at different times and by analyzing the phase differences between them. The technology of differential interferometric synthetic aperture radar (DInSAR) has become one of the most robust and economical techniques for monitoring ground displacements. It relies on the double-phase difference in time and space, known as the interferometric phase, as explained in [2]. The implemented proactive monitoring through this modern technology improves evaluation processes and reduces the risk of failure. Additionally, it raises public awareness effectively and facilitates the transition from scientific research to industrial fields.

We can obtain radar interferometry images through a group of synthetic aperture radar satellites such as Sentinel-1A/B, RADARSAT-2, COSMO-SkyMed, TerraSAR-X, and others. Satellite selection depends on the required spatial resolution, temporal frequency, waveband, and funds. In this paper, Sentinel-1 has been selected as an attractive source of information that provides satellite images of high temporal and spatial resolution. It is developed by the European Space Agency (ESA) under the Copernicus program to provide all-weather, day-and-night radar imagery on a global scale. It is equipped with a C-band, selected for its excellent balance of penetration capabilities, spatial resolution, and sensitivity to ground movements, operating at a wavelength of approximately 5.6 cm. In the realm of interferometry applications, the satellites’ interferometric wide (IW) swath mode is frequently employed, offering a ground range resolution of approximately 5 × 20 m (azimuth × range). This high resolution is pivotal for capturing detailed surface features, allowing Sentinel-1 satellite to monitor swathes of land up to 250 km wide, thus significantly enhancing the ability to monitor vast regions for ground displacements. The C-band SAR instrument supports a wide range of applications such as environmental monitoring, disaster management, and observations of land and ocean. Researchers in [3] outlined the historical development and advancements of the Copernicus Sentinel-1 mission. On the other hand, Sentinel-1’s sensor has characteristics that allow effective surface monitoring for landslides, which are relatively small in space and variable in time. Moreover, Sentinel-1 deploys the C-band spectrum that takes a short time to analyze large areas, as described by the others in [4].

DInSAR techniques can monitor ground displacements in hard-to-reach regions, contribute to developing infrastructure maintenance and management, and reduce the time, effort, and cost of traditional methods. However, they require a high level of experience to process the data and a large amount of data to obtain accurate results. Beyond these reasons, interferogram unwrapping is critical and challenging (An interferogram is a map of the phase differences between two SAR images, which is computed by cross-multiplying the first SAR image with the complex conjugate of the second, as defined in [5]). Moreover, estimating atmospheric artifacts is inevitable, selecting the reference point during processing steps is not straightforward, and even choosing the appropriate DInSAR technique can be controversial. Most of these limitations have been discussed thoroughly in the following master thesis [6].

Therefore, in this article, we discuss the need to explore less complicated methodologies to infer indications about ground displacements depending on DInSAR techniques, without addressing the above-mentioned issues. In other words, the main contribution of this article is to use intelligent algorithms in developing a methodology that can automatically analyze large InSAR data packets to identify areas where infrastructure is at risk of displacement due to ground movements. Recognizing the limitations of traditional DInSAR techniques in the context of complex data processing and the high level of expertise required, this paper delves into the application of machine learning (ML) to streamline and enhance infrastructure monitoring and risk assessment processes.

The simplicity and effectiveness of such ML-based methodologies offer a promising direction for enhancing geohazard risk assessment, as evidenced in [7], where the authors obtained 83.5% accuracy using a lightweight artificial neural network (ANN) model for validating the landslide susceptibility map in a district of Kerala in India. Recent developments in combining geographic information systems (GISs) and ML with traditional DInSAR techniques have significantly improved sophisticated risk assessments like mapping landslide-prone areas using four standard machine learning algorithms, logistic regression (LR), support vector machine (SVM), K-nearest neighbor (K-NN), and random forest (RF), as in [8]. In parallel to our work, there has been significant research on leveraging ML for landslide susceptibility mapping and utilizing ground-based synthetic aperture radar (GB-SAR) for landslide deformation monitoring. These advancements underscore the growing intersection between geospatial technologies and ML, highlighting how ML algorithms can refine the interpretation of SAR data for more effective landslide detection and monitoring. For instance, the literature review in [9] demonstrates the effectiveness of ML-based models and deep learning (DL)-based models in mapping landslide susceptibility, offering a comprehensive framework for predicting landslide-prone areas with high accuracy. Similarly, GB-SAR studies provide a detailed temporal and spatial analysis of ground movements that complement satellite-based observations, showcasing the utility of these technologies in closely monitoring landslide deformations.

Numerous articles concentrated on different aspects regarding the integration between ML and InSAR, mainly predicting the ground displacements in volcanic regions using AlexNet, a convolutional neural network (CNN), as in [10], or relying on geometrical feature engineering and advanced unsupervised learning algorithms as in [11], or by training a simple ResNet-18 convolutional neural network on the unlabeled real InSAR dataset with pseudo-labels generated from their top transformer prototype model as in [12]. Others, e.g., [13], examined an association between infrastructure displacements estimated by machine learning algorithms (MLAs) and international roughness index (IRI) values depending on persistent scatterer interferometric synthetic aperture radar (PS-InSAR) and showed that support vector machine (SVM) and boosted regression tree (BRT) algorithms are the most suitable algorithms for predicting surface movements. Clayton et al. in [14] trained labeled wrapped and unwrapped interferograms to detect the location of the surface deformation employing transfer learning, SarNet, and obtained an overall accuracy of 85.22% on a real InSAR data set. Furthermore, scholars in [15] successfully detected several types of deformation around the UK, associated with coal mining, groundwater withdrawal, landslides, and tunneling, using Synthetic examples of InSAR images and adapted a pre-trained CNN.

Hence, existing researches mainly focus on the integration between InSAR and artificial intelligence (AI) or GIS and AI. This observation has led to the idea of implementing the methodology of our research within the GIS environment.

Here, we present an innovative approach to the integration of ML, InSAR, and GIS techniques, specifically focusing on the extraction of geospatial information from wrapped interferograms using the Cosine K-nearest neighbor (K-NN) algorithm, so that our study reveals subtle spatial relationships and patterns within the data, offering a new perspective and methodology for geospatial analysis. The approach in this paper is a new contribution to the domain of ML and InSAR. This innovative method labels the pixels within wrapped interferograms based on time series data. By utilizing ML algorithms, the prediction of each pixel is then used to gather valuable information about displacements. We propose to input the wrapped interferograms with the coherence maps and check if the monitored area has a fast movement rate (positive or negative) along the line of sight (LoS) as an output.

This novel application stands out for its ability to navigate the complexities inherent in wrapped interferograms, marking a significant departure from conventional methods in the field. Another key aspect of our research is the precise formulation of datasets involving advanced preprocessing techniques, including selecting pixels based on coherence thresholds and optimal data arrangement, enabling precise extraction of geospatial insights from the complex details of interferograms. Additionally, we introduce a strategic integration of pseudo-labeling with Cosine K-NN, employing semi-supervised learning to enhance prediction accuracy. This dual-strategy approach (pseudo-labeling and Cosine K-NN), leveraging pseudo-labeling to enrich the dataset, significantly improves the model’s performance in handling complex geospatial datasets.

In summary, our research contributes to the field by offering a unique method that significantly advances ML-driven geospatial analysis of InSAR data. By combining Cosine K-NN with pseudo-labeling and innovative dataset formulation, we set new standards for accuracy and methodological approach in extracting and analyzing geospatial information. This work not only demonstrates originality and innovation but also has the potential to revolutionize the way InSAR data analysis is conducted, opening new avenues for research and practical application in the domain.

## 2. Radar Interferometry

### 2.1. Parallel Small Baseline Subset Approach (P-SBAS)

The small baseline subset approach (SBAS) is an advanced DInSAR technique that handles SAR interferograms to produce ground deformation maps and time series over large spatial scales. By utilizing data from multiple sensors, this technique can achieve outputs with millimeter-level accuracy. The combination of SBAS interferograms is characterized by a small temporal and orbital separation baseline, effectively reducing the decorrelation error associated with the interferometric phase signals, as mentioned in [16]. Despite these advantages, the SBAS technique encounters several challenges, including monitoring displacement in low-coherence areas, high computational demands, sensitivity to atmospheric conditions, and complexities in data management and interpretation, as explained by the researchers in the following review [17]. These challenges not only limit its efficiency, particularly in time-sensitive or vegetation-covered areas where temporal and spatial decorrelation can impact the accuracy of deformation measurements but also necessitate substantial computational resources and sophisticated management systems, making the process resource-intensive and less suitable for real-time applications.

The development of the parallel small baseline subset (P-SBAS) technique represents a significant advancement, addressing these limitations by enhancing processing efficiency and data management through parallel computing. By distributing the workload across multiple computing nodes, P-SBAS effectively manages large volumes of data and incorporates advanced algorithms to mitigate temporal and spatial decorrelation, thereby improving the quality of deformation measurement. This upgrade makes P-SBAS a more efficient and robust tool for monitoring surface deformations, which is crucial for disaster management, environmental monitoring, and urban planning. Furthermore, P-SBAS is designed to exploit distributed computing infrastructures, such as cloud computing environments, to optimize the usage of CPU, RAM, and I/O resources, which has been revealed through its first assessment by [18]. It implements various granularity parallelization strategies to address the most critical computation steps, significantly enhancing processing speed and facilitating near-real-time monitoring. This capability is vital for early warning systems and rapid response scenarios, supporting timely data processing and allowing for more iterative analyses and faster decision-making processes in surface deformation studies [16]. The evolution to P-SBAS, therefore, represents a leap forward in the ability to process massive amounts of data efficiently, underscoring the importance of high-performance computing (HPC) environments in surpassing the capabilities of traditional SBAS techniques, especially relevant for data from high-resolution satellite missions.

It is worth noting that further processing of Sentinel-1 Earth Observation (EO) data using the P-SBAS technique to generate wrapped or unwrapped interferograms adjusts the pixel size, or ground resolution, to approximately 26 m by 30 m. This adjustment occurs during the P-SBAS processing steps, which include multi-looking, filtering, and resampling to reduce noise and handle the data more effectively, aiming to improve the signal-to-noise ratio for a clearer detection and analysis of ground deformations. The resultant 26 × 30 m resolution, while lower than the original SAR imagery, enhances the interpretability of the interferometric products, making them more suitable for detailed geospatial analyses of ground movement. This resolution trade-off reflects a deliberate optimization between spatial coverage and the resolution necessary for precise displacement measurement, adapted to the demands of analyzing ground deformations with improved accuracy and reliability.

### 2.2. Parallel Small Baseline Subset at Geohazards TEP Platform (P-SBAS Service)

In this work, we utilize the geohazard thematic exploitation platform (G-TEP), an advanced cloud computing platform tailored to the geohazard community’s needs. G-TEP is distinguished by its innovative services, providing broad access to both optical and synthetic aperture radar (SAR) data. Cloud computing (CC) serves as a pivotal resource in the era of big data, prized for its scalable and flexible architectures, cost-effective computations compared to in-house solutions, straightforward usage, and the growing availability of public cloud environments. These attributes facilitate resource optimization and enhanced performance, as outlined in [18].

G-TEP leverages these cloud computing strengths to offer extensive satellite data access and advanced data processing tools. This allows for the efficient management of large datasets and complex analyses. The platform supports customizable workflows, enhancing productivity and fostering a collaborative environment by enabling the sharing of data and results. Designed to be user-friendly, G-TEP is accessible to a broad audience, further supported by comprehensive training and robust support resources. This combination of features makes G-TEP an invaluable tool for advancing geohazard analysis and improving disaster preparedness, response, and recovery.

Our research has significantly benefited from scalability and optimization provided by the CC, particularly in the comprehensive analysis of Sentinel-1 imagery. We successfully processed entire frames of Sentinel-1 images, generating both wrapped and unwrapped interferograms, coherence maps, and velocity deformation maps. This achievement underscores the seamless scalability of CC resources, a capability that is largely unattainable with traditional computing systems. Optimization was another area where our research benefited greatly from the CC. The analysis of deformation velocity maps extended beyond mere velocity values at each coherent pixel; it encompassed geographic coordinates, coherence, topographic phase, and displacements along the LoS. This comprehensive approach to data management exemplifies the professional handling and processing capabilities provided for our computational framework. Furthermore, enhanced collaboration, as an additional optimization aspect, was markedly useful through the utilization of the platform’s built-in tools for data sharing. In addition, the development of customized notebooks by the platform facilitated our bulk data downloading processes, streamlining our workflow.

The P-SBAS processing chain at the G-TEP encompasses: selecting appropriate S-1 IW SLC images, converting DEM into SAR coordinates and geometrical co-registration of SAR images, generating differential interferograms with Goldstein filtering, performing phase unwrapping based on coherence, estimating deformation and residual topography, generating displacement time series and velocity maps, applying atmospheric phase screen (APS) filters on time series, and finally geocoding and integrating results into GIS. A detailed study about this chain is provided by [19].

## 3. Datasets

The developed methodology was tested using three distinct datasets. In this section, we elucidate the sensitivity of each dataset to landslides and provide a rationale for its selection. Furthermore, we outline the parameters employed for the P-SBAS analysis of each dataset. The parameters for the start and end dates are thoroughly explained in the subsequent paragraphs. The number of images was determined by the start and end dates and our desire for consistency in extent and framing across the selected time frame. The temporal coherence was a key consideration, varying by location. In Milan’s urban area, high coherence levels enabled a higher coherence threshold, ensuring data reliability. Conversely, lower coherence levels in Washington required a reduced threshold. For Lisbon, the approach was guided by prior research, adapting the coherence criteria accordingly. The SRTM1 dataset, with its 1 arc-second (∼30 m) resolution, is favored for deformation monitoring due to its blend of high spatial detail and extensive global coverage, enabling precise, consistent analysis across diverse terrains and regions for free.

### 3.1. Lombardy Dataset

Lombardy is one of the most landslide-active regions in Italy (see Figure 1) as reported in the database of landslides that occurred in Italy between 1279 and 1999 by [20]. It is located in the north, where mountains represent 40.5% of its composition. In general, the most exposed areas to high or very high landslide danger are Sondrio at 14.5%, Bergamo at 12.6%, Lecco at 10.3%, Brescia at 8.3%, and Como at 8.1%, respectively, as reported by the Italian National Institute for Environmental Protection and Research (ISPRA) [21]. The most important reasons to classify Lombardy as a highly susceptible area to landslides are the long-term consumption in places that should not be used for human activities and the geomorphological factors that deem the territory (see Figure 2). Table 1 shows the selected parameters of the P-SBAS analysis for the Lombardy Dataset (The target region of the analysis was Lombardy. But using the P-SBAS service, we could obtain displacement information over other regions such as Veneto and Trentino-Alto Adige too). The time range considered in this analysis encompasses a major flood occurrence that occurred on 26–27 July 2021, affecting several towns around Lake Como like Blevio, Brienno, Laglio, Cernobbio, and Argegno. The flood caused substantial destruction to residential and commercial structures, blocked road segments, and triggered numerous landslides.

### 3.2. Lisbon Dataset

The metropolitan areas of Lisbon (LMA) in Portugal are recognized as areas of high susceptibility to landslides according to the European landslide susceptibility map (see Figure 3) and the landslide risk index (LRI) in Portugal developed by S. Pereira et al. in [22]. The exposure dimension is the main incentive of the LRI in LMA. It is related to the population density and the number of buildings. In addition, LMA experiences dangerous floods during heavy rain periods, slope instabilities, and active gravity processes along the coast and inland, which makes it prone to seismic and tectonic geological hazards [19]. A buried landslide with a length of 11 km, a width of 3.5 km, and a maximum thickness of 20 m was reported in 2019 using high-resolution multichannel seismic reflectance profiles. This landslide caused the collapse of half of the upper unit of the Tagus River delta front in the Holocene era. Scholars in [23] discussed the trigger mechanisms of this newly identified landslide. Table 2 shows the parameters chosen for the P-SBAS analysis of the Lisbon dataset (The target region of the analysis was Lisbon, but using the P-SBAS service, we could obtain displacement information over Leiria, Santarém, and Setúbal).

### 3.3. Washington Dataset

According to the U.S. landslide inventory, Washington—the western coastal state of the U.S.—is highly susceptible to landslides (see Figure 4). Yuan Kun Xu et al. in [24] confirmed this susceptibility over the entire U.S. West Coast states by identifying slow-moving, large-scaled, and active landslides (4–17 cm/year along the radar Los). Those landslides were delineated from displacement signals captured by ALOS-2 PALSAR-2 images between 2015 and 2019 (see Figure 5). Depending on extensive observations, the spatial density and size of those landslides were significantly controlled by bedrock type. On the other hand, their occurrence and continual movement revealed long-term land uplift. Table 3 shows the parameters chosen for the P-SBAS analysis of the Washington dataset. The time range chosen for this study aligns with the time frame utilized by Yuan Kun Xu et al. as a reference for the occurrence of landslides in the studied area. This selection serves two main purposes: to establish a temporal benchmark and to evaluate the effectiveness of Sentinel-1 in detecting and capturing these landslides.

## 4. Methodology

The developed methodology aims to infer a trend movement indication of the ground based on the wrapped interferograms and coherence maps. In simpler terms, the inputs are the pixels selected from the wrapped interferograms using a coherence threshold, and the outputs are the prediction results classifying each pixel into one of the following five categories using three different models: fast positive, positive, fast negative, negative, or undefined movements. The first model specifically addresses positive and negative movements. The second model is dedicated to fast positive or undefined movements. The third model is dedicated to fast negative or undefined movements. We are not proposing a binary hierarchical approach. However, if a sufficient and equal number of positive, negative, and (fast positive or fast negative) training samples exist within the same extent (a challenging situation in the same region from a seismological point of view), our framework can be executed in a binary hierarchical way. Neglecting the importance of balanced and adequate training samples can lead to several detrimental consequences for the model. Specifically, it may induce a bias toward the majority class, result in deceptive performance metrics lead to the loss of important information, and may also promote overfitting. Consequently, the adopted methodology proposes training the three models for each specified area. The model that exhibits the highest accuracy is then selected as the most reliable. The ground truth has been created based on the velocity values of the measurement points (MPs). These values were derived from the time-series results using the P-SBAS service. We opted to utilize wrapped interferograms instead of unwrapped interferograms due to the substantial computational burden associated with the unwrapping process in DInSAR analysis. Consequently, employing the wrapped interferograms to derive indications of movement yields results that offer greater advantages. In the following subsections, we provide a short description of each workflow step.

### 4.1. Producing the Wrapped Interferograms, Coherence, and Time Series Using the P-SBAS Service at the G-TEP Platform

The main reason for using the G-TEP Services is to acquire enough training examples for training the appropriate model and obtaining the final pixel predictions.

The P-SBAS approach was preferred over the PSI approach for two reasons. The first is the higher density of measurement points that P-SBAS provides. The second is the design of the P-SBAS processing chain which runs many steps in parallel by exploiting different bursts as inputs to reduce the time frame.

### 4.2. Creating the Dataset of the Selected Pixels and Corresponding Labels

The datasets for the selected interferogram pixels, along with the associated labels describing their movements, were generated using the produced time series, coherence maps, and wrapped interferograms. Initially, as the interferograms generated by P-SBAS G-TEP are not arranged chronologically, they were sorted chronologically before dataset creation and model training samples input. They were sorted in ascending order according to the dates of the slaves images (see Figure 6).

Secondly, the atmospheric artifacts are removed from the wrapped interferograms to reduce the global and local phase errors, thereby improving the accuracy of the predicted signal. This is because the atmospheric heterogeneity introduces an APS difference which poses a significant challenge to the accuracy of displacement measurements obtained through InSAR. Accordingly, we applied a high pass filter on the interferograms, considering that the main power of the low-frequency signal comes from the atmospheric artifacts as discussed by the researchers in [25] through their interferometric processing and by other researchers in [26].

In our study, we implemented a high-pass filtering process in the spatial domain, which involved employing the averaging low-pass filter by convolution. A detailed description of this filter is available in [27] to isolate low-pass components. Subsequently, these components were subtracted from all-pass components to exclusively derive the high-pass components. Considering the consistent spatial pattern observed in the APS over a distance of approximately 1 km, as described by [28], and taking into account the pixel’s spatial resolution, we determined the kernel window size. The value of the kernel window size was chosen with careful consideration to retain an appropriate balance between preserving relevant spatial information from the original image and minimizing undesirable noise.

By virtue of this operation, the methodology effectively mitigates the highlighting of displacements sharing similar wavelengths. As a result, this enhances its suitability for precisely detecting and characterizing localized phenomena. After filtering the interferograms, we decided to use the magnitude values for training the filtered interferograms (see Figure 7). This decision is based on the idea that the magnitude of a complex number (which represents the values of a wrapped interferogram in the complex domain) indicates its distance from the origin in the complex plane. For values on the unit circle, this magnitude is 1. After filtering, these values can change. Magnitudes close to 1 suggest that the high-pass filtered phase at that location is close to the original phase value. Accordingly, the filtering process did not significantly alter the phase information at that point. On the other hand, when the magnitudes are different from 1, this indicates that the high-pass filtering process has emphasized or de-emphasized the phase information at that location.

Magnitude values that notably deviate from 1 might signify regions experiencing substantial or abrupt phase changes. In a displacement scenario, these could be regions of significant ground movement. Moreover, by comparing the filtered results across different periods or events, the evolution or dynamics of the displacement phenomenon can be studied, potentially revealing trends, stabilization, or exacerbation of the ground movements. Therefore, these values are crucial for analyzing patterns that align with the training’s aim.

It is fundamental to note that the pixels selected as training samples had at least a 0.7 temporal coherence (The temporal coherence as described in [29], is a quality index used to evaluate the retrieval of the original phase in SAR data, particularly in the context of DInSAR deformation time series analysis. It is defined for each pixel based on the consistency of the phase measurements over multiple temporal acquisitions. It is calculated as the magnitude of the average of the complex exponential of the phase differences across all acquisitions in the time series, with values ranging from 0 to 1 threshold provided by the time series). We determined a temporal coherence threshold of 0.7 based on the P-SBAS service provided by G-TEP. The service requires a minimum temporal coherence of 0.7 to operate effectively. In essence, the dataset is structured to include the training samples extracted from the sorted and filtered interferograms, representing the spatial information, and the corresponding labels derived from the velocity values, providing the displacement information. This combined dataset serves as the foundation for training and further analysis in the study.

### 4.3. Detecting the Difference between the Movement Signals

Before feeding the datasets to the machine learning models, we carried out a study to differentiate between fast and slow movements. To achieve this, we extracted two matrices from datasets associated with fast movements. Subsequently, these matrices were visualized as images, with color scales automatically adjusted to their respective data ranges (For this visualization, the imagesc function in Matlab was employed, representing a matrix as an image. In this representation, distinct matrix values correspond to different colors, facilitating a clearer understanding of data distribution or patterns and allowing for the easy identification of any common trends or variations among the datasets easily). As depicted in Figure 8, the matrices of slow negative movements comprise training samples associated with velocity values ranging from 0 to −0.7 cm/year. Conversely, the matrices of fast negative movements include training samples associated with velocity values less than −0.7 cm/year. It is visually evident that the matrices representing fast movements, which are darker than the corresponding matrices of slow movements, contain a larger number of high values, which contributes to a clearer contrast between the two types of movement. Additionally, we statistically checked the slow and fast movement matrices by calculating the occurrence of magnitude values from 0 to 0.9 rad and values from 0.9 to 1 rad as in Table 4, Table 5 and Table 6. The observed numbers further validate that, for the fast movement matrices, there is a higher frequency of occurrences of magnitude values between 0 and 0.9 rad compared to those of slow movement ones. Alternatively, for the slow movement case, the pattern is reversed, with a higher frequency of magnitude values ranging from 0.9 to 1 rad. Subsequently, this aligns with the association of slow/fast velocity points with the chosen pixels within the interferograms. Analogous results were observed when matrices for slow and fast movements were derived from the datasets of positive fast movements. In the context of fast positive movements, the slow movement matrices included training samples associated with velocity values from 0 to 0.7 cm/year, while the fast movement matrices included training samples associated with velocities greater than 0.7 cm/year.

This implies that many pixels, whether moving slowly or quickly, follow distinct, consistent trends that differentiate them. However, these differences do not possess a readily apparent explanation, and it is worth noting that they vary significantly across different datasets. Hence, it would be beneficial to conduct further in-depth trials and analyze more datasets.

On the other hand, we examined positive versus negative moving pixels. We partitioned each dataset of positive/negative movements into two matrices. The first matrix included samples with velocity values greater than zero, reflecting positive movement. Conversely, the matrix designated for negative movement included samples related to velocity values less than zero. However, we observed a consistent pattern compared to the slow and fast movements, albeit less pronounced. Specifically, positive movement matrices presented a higher quantity of values between 0.9 and 1 compared to their negative movement counterparts. Consequently, this prompted a more detailed examination of the datasets by plotting histograms of the velocity values associated with the trained samples, as the observed differentiation pattern was unforeseen. The histograms revealed that large negative velocity values occur more frequently than positive velocity values as Figure 9 shows. Thus, while the model could distinguish between positive and negative movements, it categorized positive movements as slow movements and negative movements as fast movements. Therefore, we recommend developing datasets for positive and negative movements that feature identical histograms concerning their velocity values. This approach aims to discern whether the capacity to differentiate between positive and negative movements exists and to identify the accurate patterns for such differentiation.

This finding explains the challenge of multi-class classification of the datasets. In other words, it was challenging to train a single model to classify the pixel movement into one of the following classes: fast negative, negative, positive, and fast positive. To address this issue, it was necessary to explore alternative approaches, such as training separate models for fast positive movement, fast negative movement, and positive/negative movement. By focusing on each model individually, we could better capture the specific characteristics and patterns associated with different types of movement.

### 4.4. Training the Model and Obtaining the Predictions

Three different models were trained to obtain the final predictions. The first model was designed to differentiate between positively and negatively moving pixels. The second model aimed to detect if a pixel exhibits a fast negative movement rate. The third model was tasked with determining if a pixel has a fast positive movement rate. A 0 cm/year threshold was set to train the first model. If the velocity value of the MP is less than 0 cm/year, the label will be negative, and if the velocity value of the MP is greater than 0 cm/year, the label will be positive. Then, a −0.7 cm/year threshold was selected to train the second model: if the velocity value of the MP is less than −0.7 cm/year, the label will be fast negative; if the velocity value of the MP is greater than −0.7 cm/year, the label is undefined. Correspondingly, a 0.7 cm/year threshold was assigned to train the third model. If the velocity value of the MP is greater than 0.7 cm/year the label will be fast positive; if the velocity value of the MP is less than 0.7 cm/year, the label is undefined. It is worth noting that the ±0.7 cm/year threshold was determined experimentally. According to consulting InSAR scientists, a movement rate of ±0.3 cm/year is considered significant. Therefore, we initially set thresholds at ±0.3 cm/year, then ±0.5 cm/year, and finally, ±0.7 cm/year, observing that classifier performance improved at the ±0.7 cm/year threshold. We did not explore thresholds higher than ±0.7 cm/year to balance achieving sensitivity in detection and obtaining reliable results.

In the same context, the developed datasets present an exceptional and complex nature of data. Many scenarios, such as the noise caused by atmospheric effects, the density of MPs, or the specific type of InSAR images, could reduce the overall accuracy. Experiencing several machine learning methods, we found that the Cosine K-NN model achieved the best test accuracy (These methods include Cosine K-NN, subspace K-NN (ensemble), medium neural network, logistic regression, cubic SVM, medium tree, fine tree, bagged tree (ensemble), quadratic discriminant, 2D CNN, and long short-term memory (LSTM). The machine learning models were mostly trained within Matlab. However, for the 2D CNN and LSTM models, we created specific Python codes to train them) (See Appendix A). This model is controlled by the K-NN search technique, which is a simple yet effective non-parametric supervised learning algorithm. It is extensively applied in pattern recognition due to the strong generalization feature it has, as mentioned in [30]. K-NN relies on approximation and distance functions to classify a specific data point, based on its proximity to neighboring points. The algorithm operates under the assumption of a predetermined value, k, representing the number of nearest neighbors to consider. The algorithmic target is to find the k nearest data points from a training dataset according to the closest distances from the query point. Finally, it performs a majority voting rule to determine the class that appeared the most which will be the ultimate classification for the query. These steps have been explained in [31].

The Cosine similarity K-NN model measures the similarity between two vectors *x* and *y* in multidimensional space. It detects whether the two vectors point in the same direction and operates entirely on the cosine principles, as described by [32]. The similarity between two vectors *x* and *y*, which has a range of values between 0 and 1, is calculated as in [33]:$$cos\left(x,y\right)=\frac{x\cdot y}{∥x∥∥y∥}$$

When the two vectors are closer to being collinear, the value is closer to one, and the similarity of the vectors increases. When the two vectors are closer to being perpendicular, the value is closer to zero. This means the similarity of the vectors decreases. As might be expected, this classifier achieved the highest test accuracy for most of the created datasets because its major use is to discover the correlations of vectors and specify the degree of relevance to the different classes. Thus, the main principle of how this classifier works is exactly the aim of the training process.

For more details about the principles of Cosine K-NN, the process begins with computing cosine similarity between sample vectors in a matrix S, which includes n samples and m pixel values derived from wrapped interferograms. Each row in the matrix is viewed as a sample vector within an m-dimensional feature space. The computation, as in the previous equation, is designed to assess the similarity between each pair of sample vectors. For the classification of a new sample of interest, the classifier calculates the cosine similarity between the new sample and all samples in the training dataset, arranging them from most similar to least similar. It then selects the K most similar samples, with K set to 10 as a standard starting point, to determine the classification of the new sample based on the predominant class among these top K nearest neighbors. The choice of K=10 as the starting point is made to strike a balance between overfitting on smaller datasets and underfitting on larger ones, determining how a new sample is classified based on the majority vote from its *K* nearest neighbors. The calculated cosine similarity scores, which reflect levels of similarity (where higher scores indicate greater similarity), help in the effective classification of samples based on their similarities.

The Cosine K-NN classifier’s strength lies in generalization—leveraging patterns learned during training that depend on angular relationships between vectors. An important implication of the Cosine K-NN classifier’s success is its prioritization of the angle between data points in high-dimensional spaces. This approach addresses the “curse of dimensionality” by focusing on feature relationships rather than distances, proving effective for datasets where feature interrelations are more significant than magnitude or scale, as described in [34]. Additionally, the classifier excels in handling noise and minor data variations, as described by the authors in [35], enabling accurate class distinction in interferograms despite fluctuations from noise or inconsistencies. The Cosine K-NN classifier effectively identifies complex, non-linear patterns in datasets, as mentioned in [36], making it particularly adept at distinguishing classes in interferograms with subtle variations.

The training process resulted in models with varying levels of validation accuracy (see Section 5.2). When the validation accuracy, assessed using a 5-fold validation approach, was found to be low, the pseudo-labeling technique was employed to enhance the model’s performance. Pseudo-labeling, a semi-supervised learning technique, leverages unlabeled data alongside labeled data to enhance model performance. It effectively increases the amount of data the model can learn from. This is especially beneficial in domains with scarce or expensive labeled data. By training on labeled and pseudo-labeled data, models can often achieve higher accuracy, especially in complex tasks or those with limited labeled data. This is because the model can learn more general patterns and reduce overfitting to the small set of labeled examples. It involves assigning temporary labels, or pseudo-labels, to the unlabeled data based on predictions made by a pre-trained model, as explained by [37]. It is important to note that pseudo-labeling relies on the assumption that the model’s predictions on the unlabeled data are reliable. If the model’s predictions are consistently inaccurate, the pseudo-labels may be unreliable and could adversely affect the training process. The methodology of pseudo-labeling typically involves several steps discussed by the scholars in [38]:First, a model is trained on the labeled data, using the available ground truth labels to learn the underlying patterns and relationships within the data.The trained model is then used to make predictions on the unlabeled data, assigning pseudo-labels to the unlabeled samples.Testing samples for which the model’s predictions were made with an accuracy of 0.9 or higher were combined with the original labeled data to create a new training set. We set a high-confidence threshold to ensure that only highly reliable predictions are used for generating pseudo-labels, thereby reducing the risk of compromising model performance with inaccurate labels.The pseudo-labeled training samples are removed from the original test set of unlabeled training samples.The model is retrained using the combined labeled and pseudo-labeled data, updating its parameters and improving its performance.

This process of training and pseudo-labeling can be repeated for multiple iterations until the model’s performance reaches a desired level or convergence is achieved.

In this paper, the unlabeled test samples were selected based on a 0.7 coherence threshold extracted from the average coherence map. Pixels overlapped due to the temporal coherence threshold of 0.7 set for the trained samples and the similar 0.7 threshold for the unlabeled test samples using the G-TEP overall coherence map. Some pixels exhibited both high temporal coherence and average coherence. To address the issue of overlapping pixels, an intersection is identified between the pixels selected using the threshold of average coherence and those using temporal coherence. The intersecting pixels are then removed from the testing samples selected based on the average coherence, ensuring the integrity and accuracy of the training dataset. Selecting a coherence threshold of 0.7 aims to include only those pixels that display consistent phase information over time, thereby ensuring the integrity of our training and testing datasets by avoiding data points affected by atmospheric noise or other distortive effects.

By integrating pseudo-labeled data into our models, we enabled them to learn from examples that were initially unlabeled. Our experiments demonstrate that this approach of pseudo-labeling enhances the prediction accuracy by more than 10%. The improvement can be attributed to the models acquiring knowledge about the test sets, which are closely related to the training sets in terms of geographic extent. Specifically, the test sets comprised samples not only from the same geographic area as the training sets but also from adjacent areas. To assess the model’s performance in a more granular manner, we evaluated it using unlabeled pixels scattered across the study areas. These pixels were chosen for their high average coherence within their respective regions. As a result, we achieved a prediction accuracy rate of 0.9 for a considerable number of pixels. The findings, detailed in Section 5.2, indicate that the model possesses robust generalization capabilities. In future research, we plan to explore how far the model can deviate from the training samples while still maintaining its ability to generalize effectively.

### 4.5. Visualizing and Integrating the Results in ArcGIS Environment

We extracted new datasets from the main datasets to visualize the prediction results and compare them to the ground truth within the ArcGIS environment. These datasets included ground truth labels, predicted labels, and longitude and latitude values of the MPs (Longitude and latitude values of the MPs were extracted from the time series). We compared the ground truth to the predictions in two steps. The first was to apply the Kriging method to interpolate the predictions and the ground truth. The second was to extract the sensitivity rate of the main roads to fast movements (positive or negative), by masking these roads in the interpolated raster.

The selection of ordinary Kriging (OK) as our interpolation method is motivated by the presence of spatial correlation within the dataset. This correlation, combined with the uneven distribution of our big input data, necessitates a method that can effectively estimate values at unsampled locations. OK addresses these challenges, as mentioned in [39], by incorporating the spatial relationships within the dataset, enabling us to generate reliable estimates and enhance the spatial representation of our study area. Moreover, Kriging is frequently employed in soil and geology prediction due to its recognized superior performance compared to the inverse distance weighting (IDW) interpolation method. Studies have consistently shown that Kriging exhibits lower prediction errors, making it a preferred choice in these fields, as demonstrated in [40], where the study’s findings indicated that OK surpassed IDW in the analysis of topsoil at electronic waste recycling sites in Cameroon.

The proposed solution is notably complex due to its multifaceted approach.

The initial phase involves training and prediction with the Cosine K-NN model, where complexity arises from data preprocessing, model training, and prediction processes. The K-NN model’s complexity scales with the number of training samples, making distance calculations computationally intensive.The pseudo-labeling technique further compounds complexity by requiring iterative retraining of models with labeled and pseudo-labeled data, increasing the computational load due to repeated training phases and prediction on unlabeled data.The application of real case studies from three different regions in the world emphasizes the complexity of the solution. Each study is distinct, characterized by its unique features, necessitating the development of customized techniques for InSAR data preprocessing.The integration of the results within the ArcGIS for visualizing the predictions adds another layer of complexity, involving managing large spatial datasets, performing Kriging interpolation, and executing spatial analysis tasks such as masking regions based on sensitivity rates.

Despite its computational complexity, the proposed solution offers substantial benefits by carefully selecting and integrating specific computational innovations designed to address the unique challenges of processing high-dimensional InSAR datasets and facilitating sophisticated spatial analysis. The Cosine K-NN model is strategically used for its efficiency in managing the complexities of high-dimensional data. Unlike traditional models, it leverages the orientation and directionality of data vectors, effectively sidestepping the curse of dimensionality that often hampers the analysis of complex datasets. This focus on angular similarity rather than absolute distance allows for more refined and accurate predictions of ground movement, showcasing the model’s direct applicability to the specific data characteristics of Sentinel-1 sensor output.

Furthermore, the incorporation of pseudo-labeling significantly enhances the training dataset, giving the model enhanced robustness and adaptability. This method wisely exploits high-confidence predictions to iteratively enrich the model’s learning, a critical step in overcoming the common challenge of limited labeled data in remote sensing applications. By expanding the model’s exposure to a broader array of data variations, pseudo-labeling ensures a more generalized and resilient predictive capability.

Lastly, the integration with GIS technologies for visualization is not merely a choice for enhanced data representation but a deliberate decision to enable detailed spatial analysis and practical application of the predictive insights. This integration allows for the precise examination of ground movement impacts on infrastructure and urban planning, leveraging the spatial computing capabilities of GIS to translate complex data predictions into practical insights. The rationale behind these selected computing innovations is deeply rooted in their collective ability to not only navigate the complexities inherent in high-dimensional satellite data but also to unlock valuable insights for proactive disaster management and environmental monitoring, ultimately optimizing resource use and enhancing the solution’s analytical depth.

## 5. Results

### 5.1. The Results of the P-SBAS Analyses

We produced the time series, the corresponding interferograms (wrapped and unwrapped), and coherence maps for every case study depending on the P-SBAS technique service at the G-TEP (Lisbon time series was generated by a separate researcher. This dataset served as the foundation for our further analysis and findings). The results from time series analysis provided the following information for each measurement point: latitude, longitude, coherence, velocity, topographic phase, components of LoS unit vector along the North, East, and Vertical directions, and the displacements along the LoS.

#### 5.1.1. Deformation Velocity Map of Lombardy Dataset

Figure 10 shows the deformation velocity map of monitoring Lombardy from 7 January 2020 to 17 August 2021. The velocity of displacements along LoS ranged from −4 cm/year to +4 cm/year. The mean velocity value was −0.1128 cm/year, with a standard deviation of 0.2323 cm/year. Unfortunately, we could not determine the exact location of the damage caused by the flood around Lake Como in July 2021. However, we could observe moderate displacement velocities in the towns there.

We identified areas of high sensitivity to small-scale landslides in the Belluno region (see Figure 11). We obtained this result while selecting the optimal analysis parameters for this dataset, where the experiment included Belluno and used 31 Sentinel-1 images instead of 50 images. Landslides in Belluno region have diverse patterns (translational landslide, shallow landslide, water inflows on the slope associated with widespread runoffs, solid transport, falling rocks, rotational slip, major debris flow, water outflow with solid transport), as indicated by the scholars in [41] who developed this landslides data set in 2018–2021. However, the validation accuracy for the Belluno data set was much less than the validation accuracy for the mentioned dataset in Section 3.1. Therefore, the greater the number of Sentinel-1 images, the higher the validation accuracy.

#### 5.1.2. Deformation Velocity Map of Lisbon Dataset

Figure 12 shows the deformation velocity map of monitoring LMA from 26 January 2018 to 27 April 2020. The velocity of displacements along LoS ranged from −3 cm/year to +3 cm/year. The mean velocity value was −0.1984 cm/year, with a standard deviation of 0.2938 cm/year. This deformation map analysis, conducted by other researchers and publicly available as indicated in [42], allowed us to use their parameters for generating wrapped and unwrapped interferograms and coherence maps [19]. Based on the deformation velocity map, a significant portion of the area exhibited a negative trend in movement. Moreover, occurrences of fast negative movement have been observed in various areas. The details of these areas can be observed more precisely by magnifying their specific locations by zooming in. The positively moving pixels were rare and sparse.

#### 5.1.3. Deformation Velocity Map of Washington Dataset

Figure 13 shows the deformation velocity map of monitoring Washington State from 14 October 2016 to 28 December 2019. The velocity of displacements along LoS ranged from −4 cm/year to +4 cm/year. The mean velocity value was −0.0665 cm/year, with a standard deviation of 0.4143 cm/year.

The resulting deformation velocity map of Sentinel-1 matched the uplifting and slow-moving landslides captured by ALOS-2 from 2015 to 2019 [24] (see Figure 14). The radar of Sentinel-1 explores deep-seated landslides too. Most of the slow-moving landslide locations were on the slopes of the hills where recent uplift occurred. This uplifting phenomenon, occurring over geological timescales, promotes topographical relief through increased precipitation, which in turn leads to stream downcutting and hillslope instability.

Additionally, we found that the area of postal code 98,944 has a fast negative displacement velocity according to the deformation velocity map (Figure 15) and simultaneously a high sensitivity to landslides according to the U.S. Landslide Inventory Web Application (see Figure 16).

### 5.2. The Results of the Trained Models

Table 7, Table 8, Table 9, Table 10, Table 11, Table 12, Table 13, Table 14 and Table 15 and the confusion matrices (Figure 17, Figure 18 and Figure 19) show the results of validating and testing three different models. The first model is to classify pixels into positive and negative movement classes. The second model is to distinguish pixels with fast negative movement from other pixels. The third model is to distinguish pixels with fast positive movement from other pixels. We conducted several experiments to classify pixels into fast positive, moderate, and fast negative using a single model, but the validation results were not so good. For Table 7, Table 8, Table 9, Table 10, Table 11, Table 12, Table 13 and Table 14 it was necessary to use Pseudo-Labeling (PS). However, for Table 15, the validation accuracy was better, so PL was not needed.

It is worth noting that the developed datasets were balanced relative to the number of training samples for each class, and a cross-validation of 5 folds was implemented to validate each model. Thus, the high testing accuracy indicates that our machine learning models are reliable, demonstrating strong generalization from training to unseen data. Confusion matrices in Figure 17, Figure 18 and Figure 19 provide additional evidence of the models’ reliability. This reliability is further supported by diverse and representative testing data in Figure 20 and Figure 21 of the next subsection, where test sets cover the entire studied area rather than specific parts. Moreover, considering the task’s complexity and comparisons to human performance, our models’ high accuracy proves to be sufficient, surpassing what human experts can achieve.

Due to the close proximity of measurement points in the training sets, the Cosine K-NN classifier encountered difficulties in distinguishing between their respective classes. Our pseudo-labeling technique, however, could significantly improve the prediction performance. The scarcity of training samples with velocities greater than 0.7 cm/year limited our ability to train models for predicting fast positive movement. In this aspect, pseudo-labeling offered an additional advantage by incorporating more data points into the training process based on the overall coherence of the pixel. In our analysis, we employed a coherence threshold of 0.7, which determined the inclusion of measurement points. However, lowering this threshold could increase training points and enhance our understanding of the area’s seismology by providing more information. The decision to adhere to a coherence threshold of 0.7 was justified based on its sufficiency for the provided satisfactory results.

### 5.3. Comparing the Ground Truth and the Predictions within the ArcGIS Environment

The testing sets and their predictions have been exported to ArcGIS to better visualize the results. Each MP has an assigned color according to its label. Figure 20 and Figure 21 compare the ground truth and the predictions for the three trained models of Lombardy, Lisbon, and Washington datasets respectively. After that, we interpolated the predictions and the ground truth. Finally, we masked the important roads (primary, secondary, road, motorway, and bridleway) over the studied areas of Lombardy, Lisbon, and Washington. Figure 22, Figure 23 and Figure 24 compare the masked roads that pass across the test set and their predictions for the three trained models of the datasets.

These figures clearly show only minor differences between the ground truth and the predictions for the roads in the test sets. Thus, wrapped interferograms and coherence maps provide an adequate and accurate indication of infrastructure displacements.

## 6. Limitations and Future Work

The limitations identified in our study can be summarized as follows:Our analysis differentiated between pixels exhibiting positive and negative movements. However, analysis of the histograms for these datasets revealed that this distinction was primarily based on the velocity of movement, being either fast or slow. Future work should aim to refine these datasets to address this issue and to develop a deeper understanding of the patterns that differentiate positive from negative movements. This limitation does not affect the reliability of the results but such refinement is necessary to improve the accuracy and utility of the datasets.Using a single model, it was challenging to classify three distinct categories (positive/negative, fast positive/undefined, and fast negative/undefined). Thus, we trained three separate models. Neither does this limitation affect the reliability of the results.We observed that the distribution of measurement points across the study area plays a crucial role, with denser clusters of points leading to challenges in prediction accuracy. Future research should explore how the spatial distribution of these points affects predictive accuracy, as this understanding could inform new strategies for optimizing data collection and analysis, thereby enhancing the precision of predictive models. This limitation does not compromise the reliability of the results. However, by analyzing the spatial distribution, we anticipate a significant improvement in accuracy, thereby minimizing the necessity for pseudo-labeling.We employed a high-pass filter to reduce atmospheric artifacts in the interferograms. While this approach did not directly tackle low-pass atmospheric components, their influence on data quality and accuracy is considered minimal. This is because the significant atmospheric disturbances, which are typically found in the high-frequency range, were effectively mitigated, rendering the impact of low-pass components negligible.

In summary, we acknowledge the existence of limitations, yet demonstrate that they do not detract from the overall validity and robustness of our results.

## 7. Conclusions

Through the utilization of filtered wrapped interferograms and coherence maps, we successfully developed datasets that robustly identified indications of ground displacements.

Numerous machine learning methods were tested to achieve this goal. Cosine K-NN demonstrated the highest accuracy in determining whether the pixel had a fast (positive or negative) movement rate, particularly for pixels within adjacent geographic regions. We trained three different models (positive/negative, fast positive/undefined, and fast negative/undefined). This approach was necessitated due to the dataset’s limitations in simultaneously predicting all three classes with a single model.

In response to the challenge of low validation accuracy, we successfully implemented pseudo-labeling as a strategy to overcome this obstacle. By incorporating pseudo-labeling into the training process, we observed significant improvements in the accuracy of the models and gained a deeper insight into the displacements of the studied area by collecting additional training samples. We focused on the main roads to predict their sensitivity to fast positive or negative movement. Given the ability to assess the movement status of individual pixels, it follows intuitively that we can extract indications regarding adjacent structural elements within the area, extending beyond the scope of roads alone.

The careful selection and initial assignment of the used parameters, are achieved through a mix of experimental adjustments, expert consultation, and data-driven considerations.

Essentially, the processing of Sentinel-1 sensor data for ground movement detection utilized advanced machine learning techniques, notably the Cosine similarity K-NN model, along with GIS integration. Focused on the angular relationships between vectors, the Cosine K-NN model excelled in analyzing the complex patterns in InSAR data, proving indispensable for accurately identifying and classifying ground movement phenomena. Pseudo-labeling was employed to enhance the training dataset, effectively addressing the scarcity of labeled data and improving model generalization. The integration of model predictions with ArcGIS for spatial analysis and visualization was crucial, enabling an in-depth comparison of predictions with ground truth and facilitating the evaluation of geohazard impacts on infrastructure. These computing innovations collectively elevated the efficiency and accuracy of environmental monitoring using Sentinel-1 data, striking a balance between computational demands and advanced analytical capabilities for geohazard monitoring and disaster management.

Moreover, we utilized three distinct datasets from various geographical regions and discerned a consistent pattern across all of them. However, this observation should not be considered unequivocal and requires further investigation. Additional investigation is requisite to validate and fortify these findings.

Nonetheless, based on our extensive experience in this research, we conclude that the deformation signal can be directly extracted from the wrapped interferograms and coherence maps with the aid of machine learning.

The deployment of machine learning in this process has been instrumental in substantiating our conclusion and enhancing our understanding of the underlying phenomena. Furthermore, we chose not to merge the models from three different datasets into a single, unified model, primarily due to our focus on three distinct regions. Each of these regions has its own unique attributes and deformation patterns, which cannot be generalized or confined to other areas, or even to the same regions across different time spans. Consequently, there was no need to input the temporal baselines of the interferograms or establish other specifications during the training process. Instead, we have chosen an approach that allows for each region’s dataset to be recreated and its model to be retrained individually. This ensures that the results are specific to each particular area and also enables the potential for this approach to be applied in various regions in future applications. Finally, we highly recommend evaluating the outcomes obtained by implementing the persistent scatterer interferometry technique (PSI) using the same workflow.

## Figures and Tables

**Figure 1 sensors-24-02637-f001:**
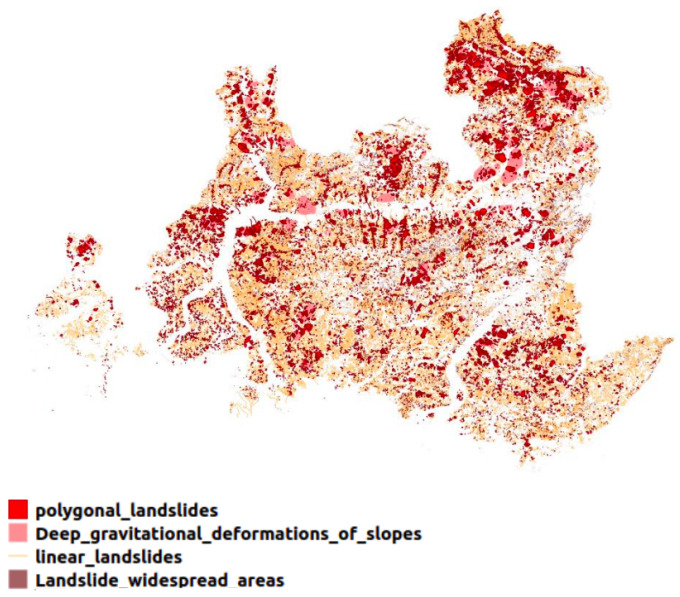
Landslide distribution in Lombardy (Italy) according to the Geoportal of Lombardy.

**Figure 2 sensors-24-02637-f002:**
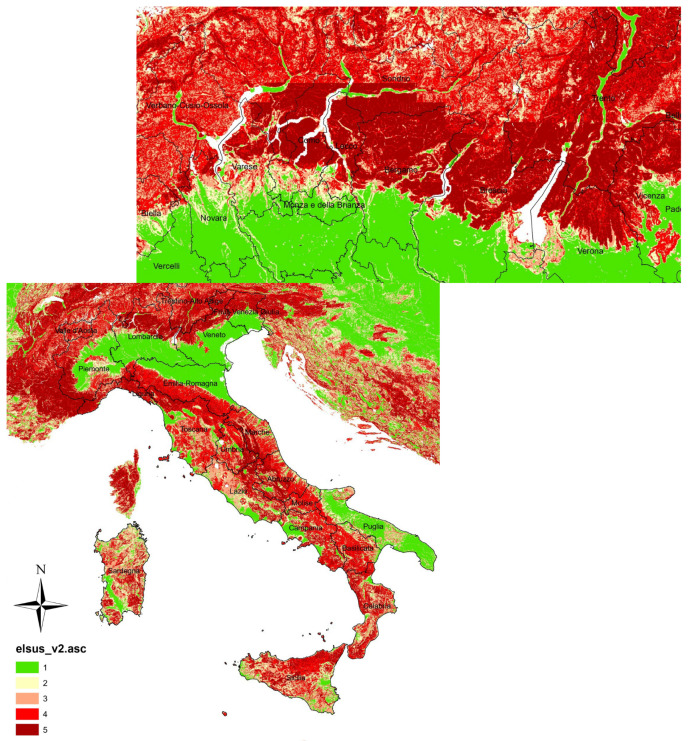
European landslide susceptibility map (Italy, Lombardy) according to the European Soil Data Center. The colors from green to red represent the sensitivity degrees from low to high.

**Figure 3 sensors-24-02637-f003:**
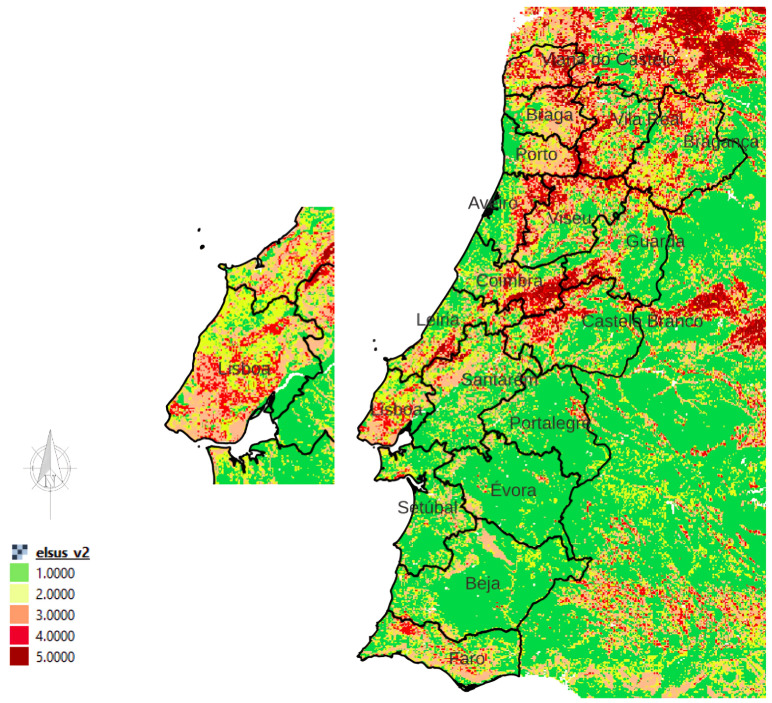
European landslide susceptibility map (Portugal, Lisbon) according to the European Soil Data Center. The colors from green to red represent the sensitivity degrees from low to high.

**Figure 4 sensors-24-02637-f004:**
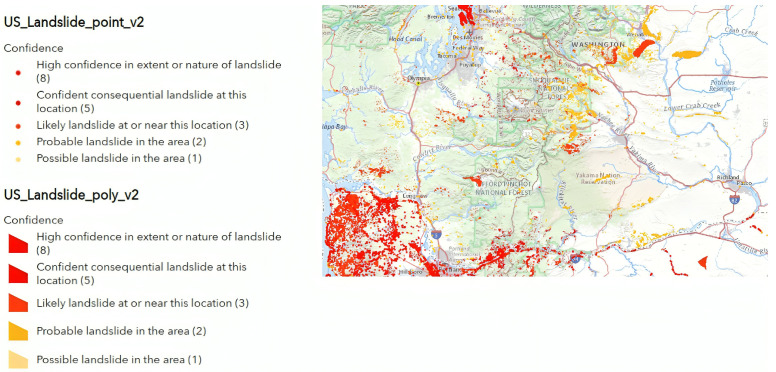
Washington landslide susceptibility map according to the United States geological survey.

**Figure 5 sensors-24-02637-f005:**
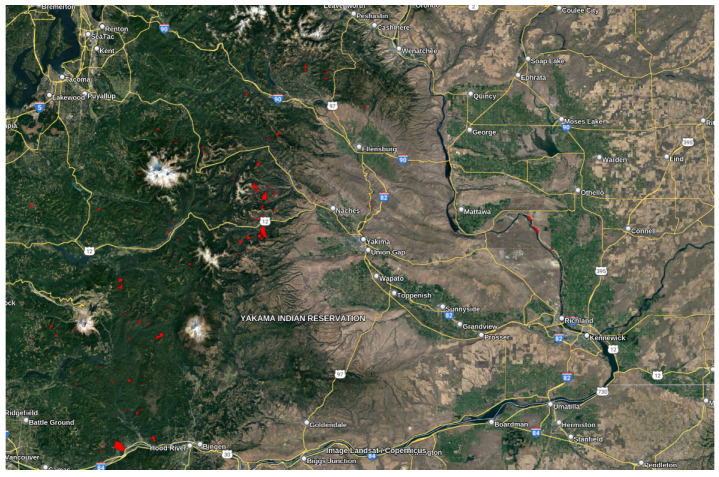
Washington deep-seated landslides captured by ALOS-2 PALSAR-2 images between 2015 and 2019. The red polygons represent the landslide locations.

**Figure 6 sensors-24-02637-f006:**
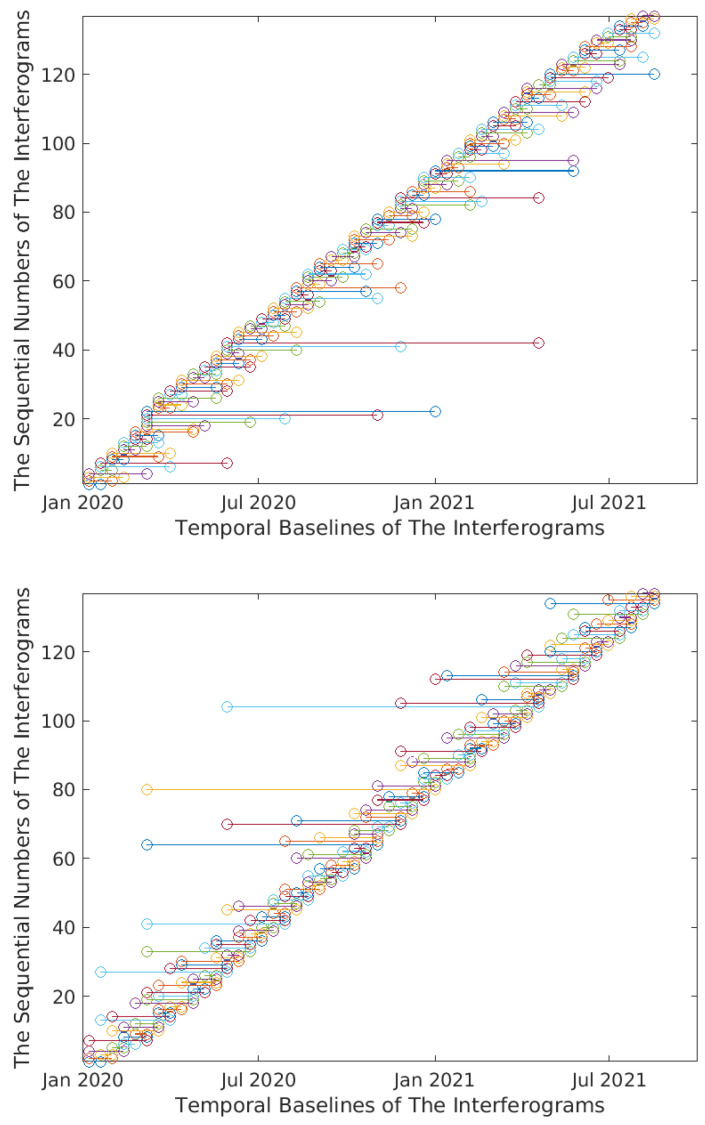
The chronological sorting of the temporal baselines of the wrapped interferograms (Lombardy Dataset). **Top** figure expresses the sequence of the temporal baselines of the interferograms before sorting. **Bottom** figure expresses the sequence of the temporal baselines of the interferograms after sorting. Similar results have been obtained for the other two datasets of Lisbon and Washington.

**Figure 7 sensors-24-02637-f007:**
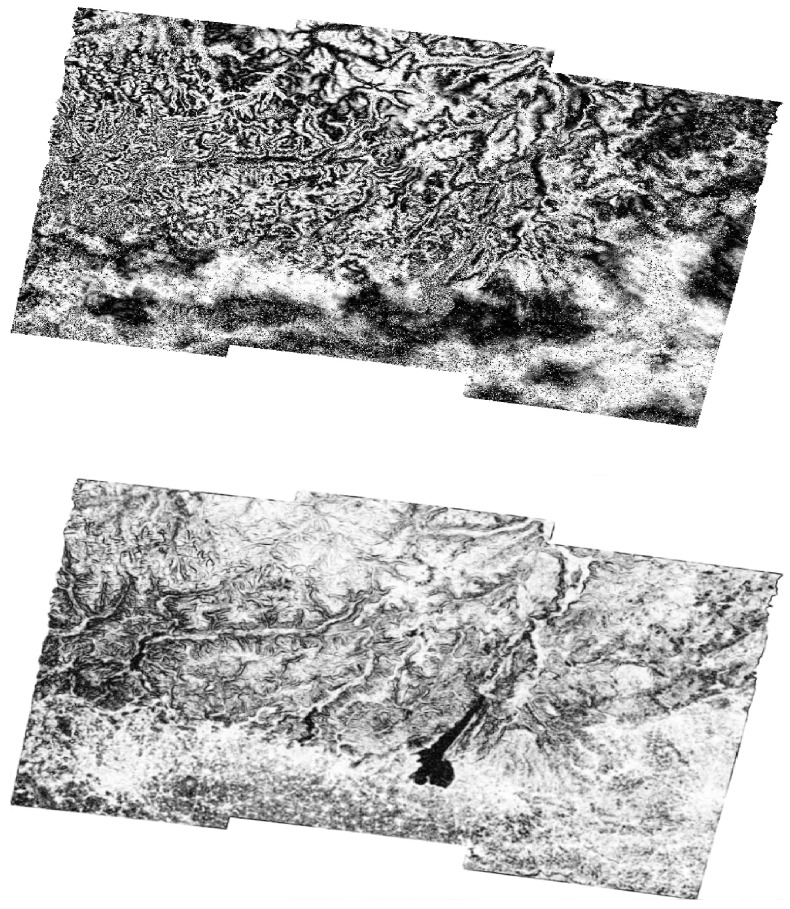
Example of a wrapped interferogram in the complex domain from the Lombardy dataset before using the high-pass filter (**top** figure) and the magnitude after using the high-pass filter (**bottom** figure).

**Figure 8 sensors-24-02637-f008:**
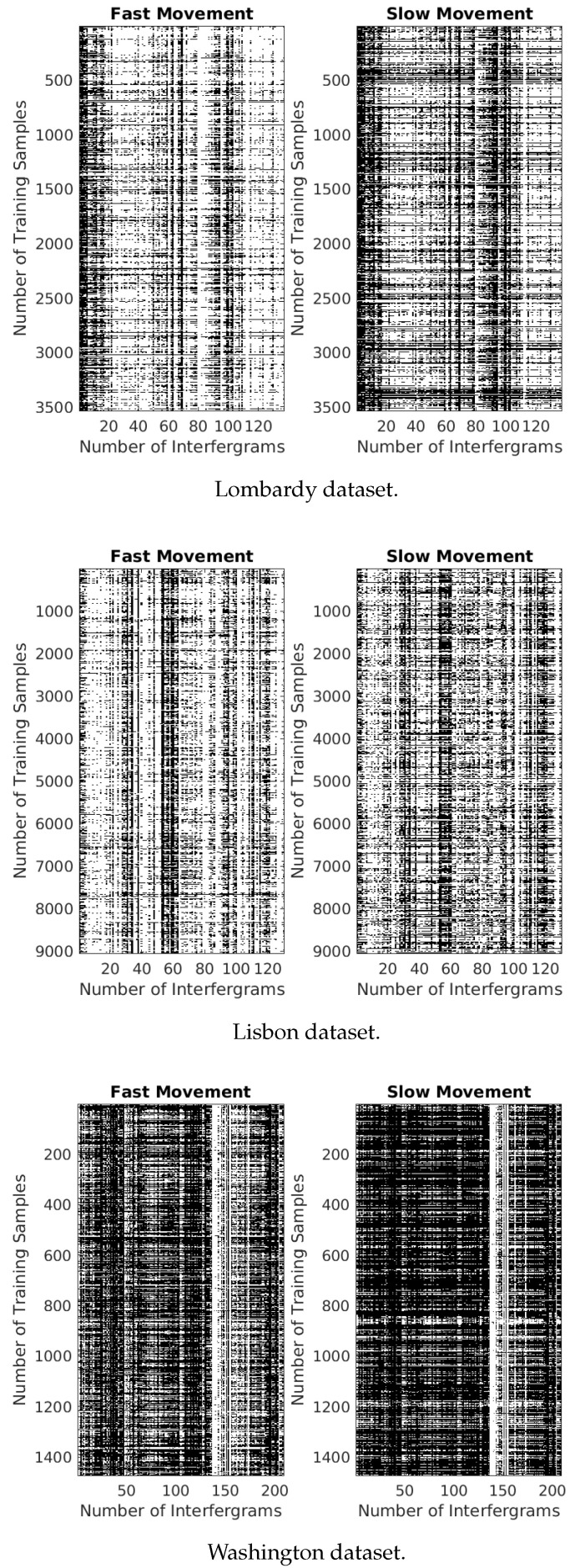
The matrices represent slow and fast motions based on the used datasets. The black color in the matrices represents the magnitude values greater than 0.9 radians, while the white color represents the filtered phase values smaller than 0.9 radians.

**Figure 9 sensors-24-02637-f009:**
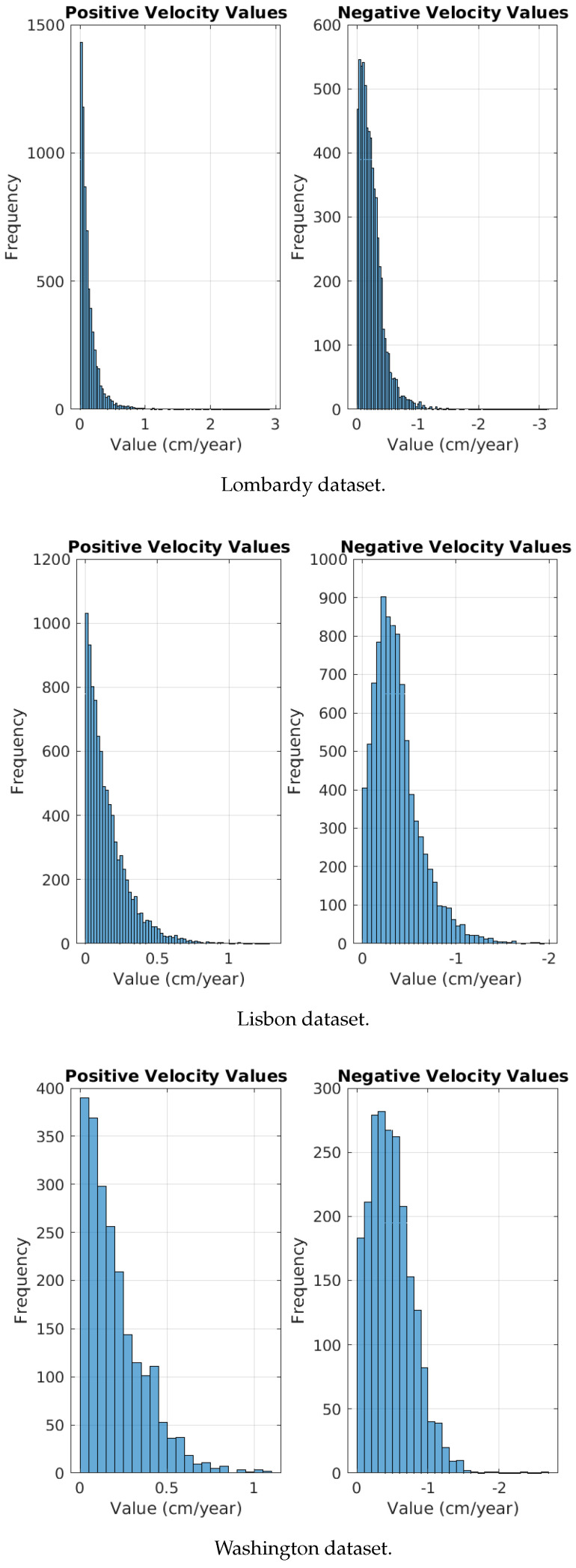
The histograms representing positive and negative motions based on the used datasets.

**Figure 10 sensors-24-02637-f010:**
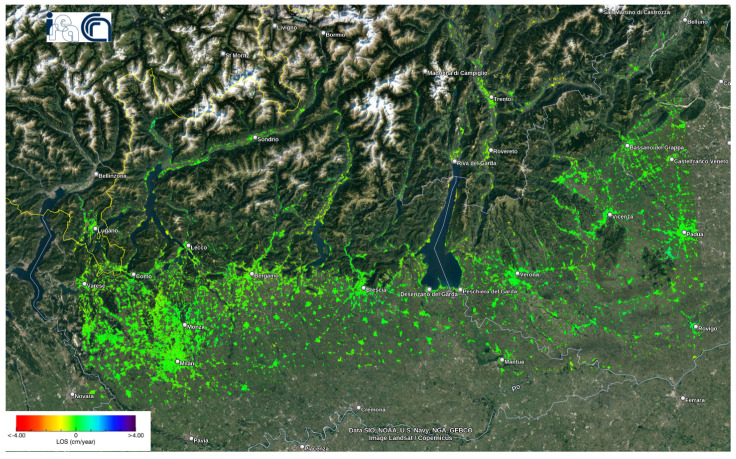
Deformation orm as it is so keep velocity map of the Lombardy dataset using the P-SBAS service at the G-TEP.

**Figure 11 sensors-24-02637-f011:**
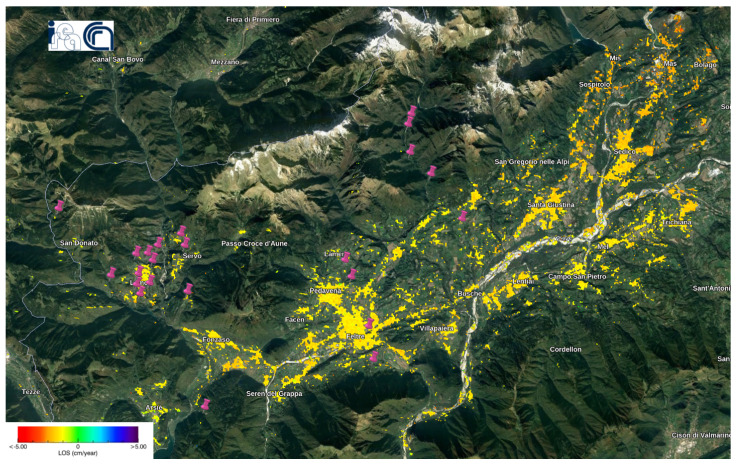
Intersection between S.Puliero et al.’s landslide dataset and the Sentinel-1 deformation velocity map in Belluno [41]. The violet pins refer to the location of the landslides.

**Figure 12 sensors-24-02637-f012:**
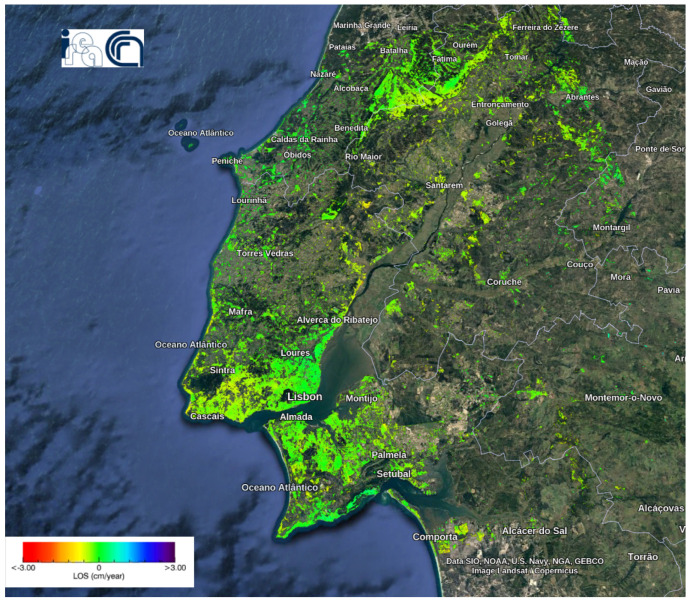
Deformation velocity map of the Lisbon dataset using the P-SBAS service at the G-TEP.

**Figure 13 sensors-24-02637-f013:**
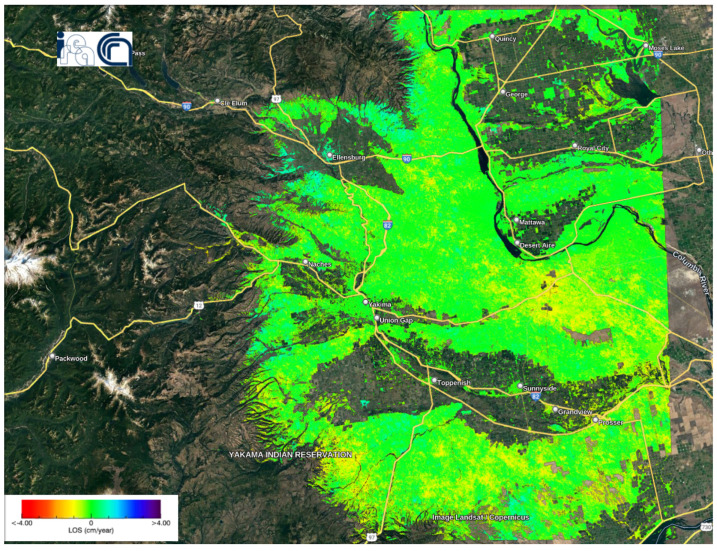
Deformation velocity map of the Washington dataset using the P-SBAS service at the G-TEP.

**Figure 14 sensors-24-02637-f014:**
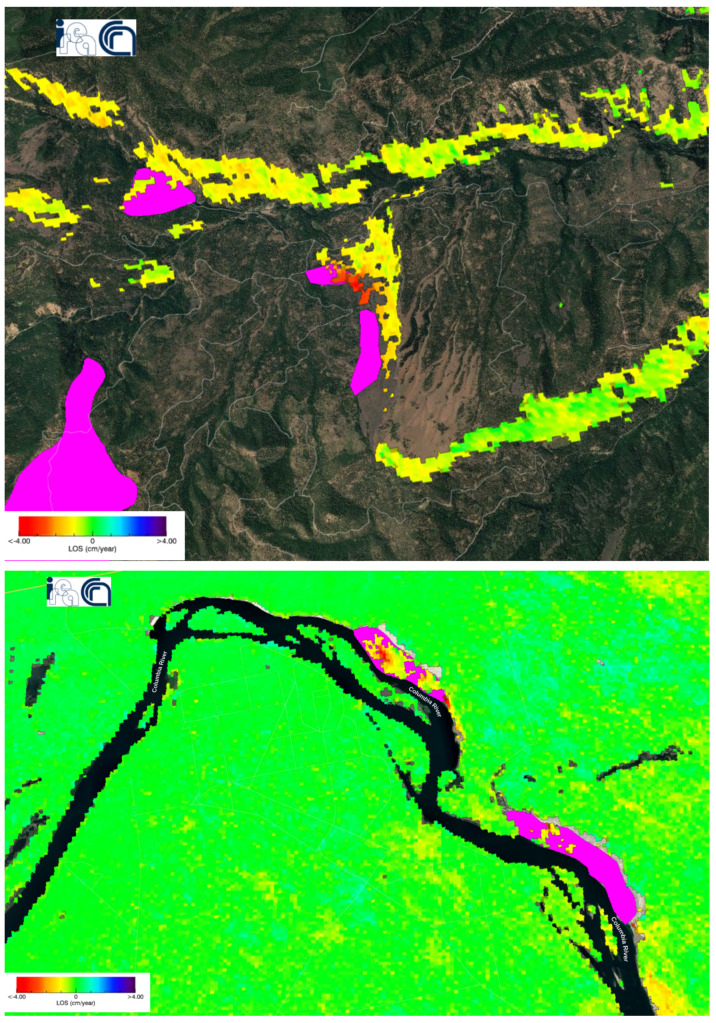
Intersections between landslides dataset [24] and deformation velocity map in the Washington U.S. The violet polygons represent the landslides dataset.

**Figure 15 sensors-24-02637-f015:**
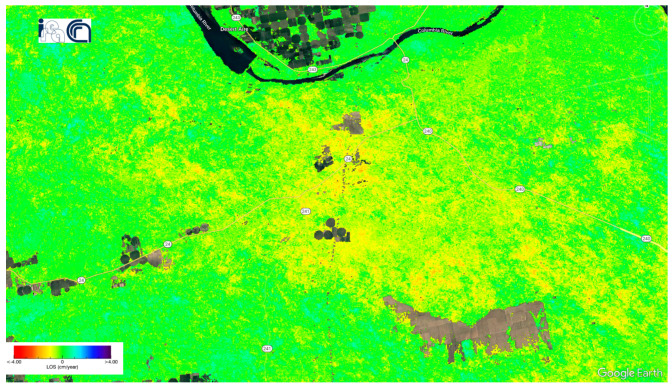
Deformation velocity map of zone 98,944 according to the time series analysis of the Washington dataset.

**Figure 16 sensors-24-02637-f016:**
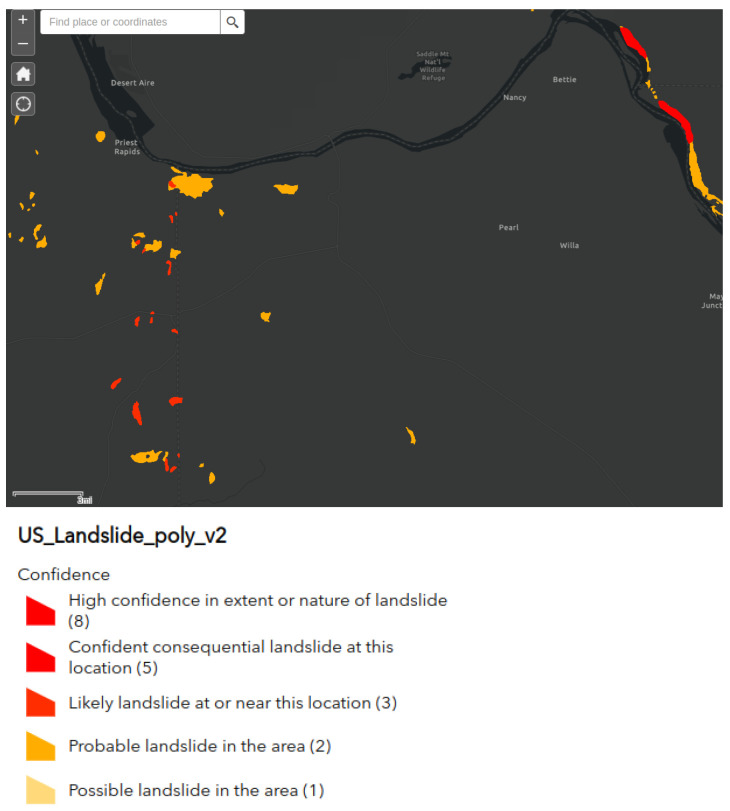
Sensitivity map to landslides in zone 98,944 according to the U.S. Landslide Inventory Web Application.

**Figure 17 sensors-24-02637-f017:**
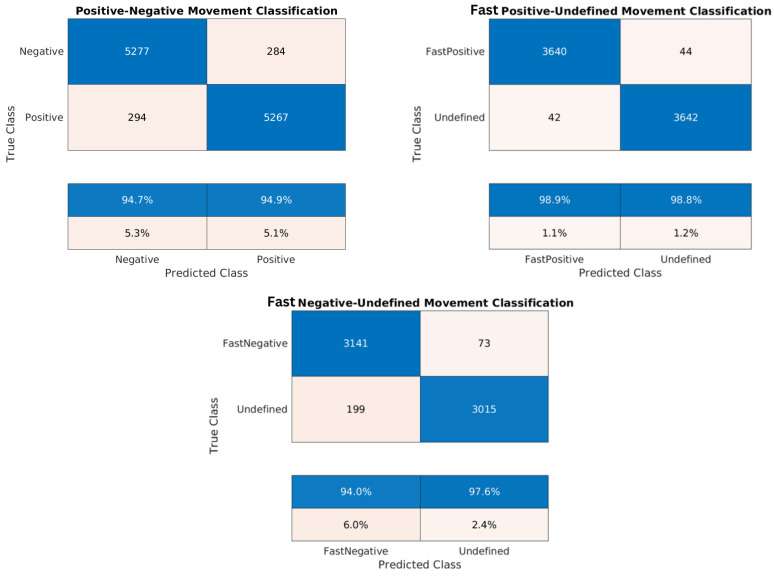
Confusion matrices for the trained models of the Lombardy dataset: positive/negative movement model, fast positive movement model, and fast negative movement model, respectively.

**Figure 18 sensors-24-02637-f018:**
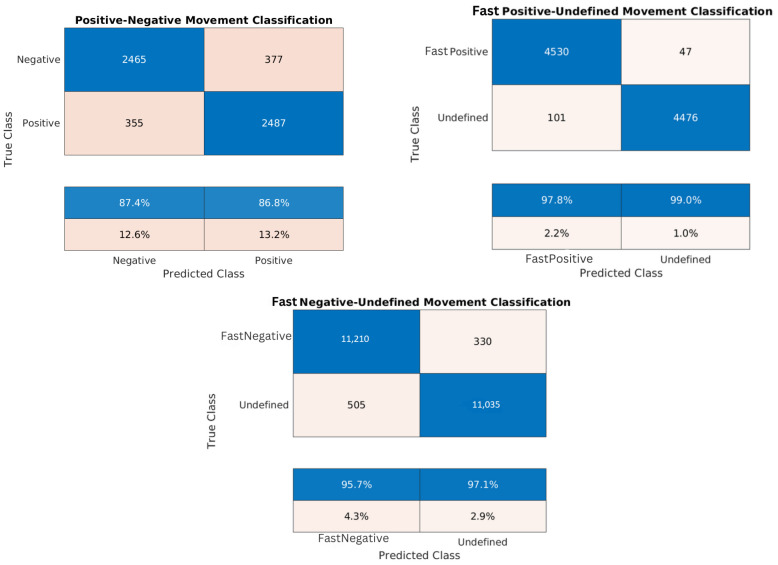
Confusion matrices for the trained models of the Lisbon dataset: positive/negative movement model, fast positive movement model, and fast negative movement model, respectively.

**Figure 19 sensors-24-02637-f019:**
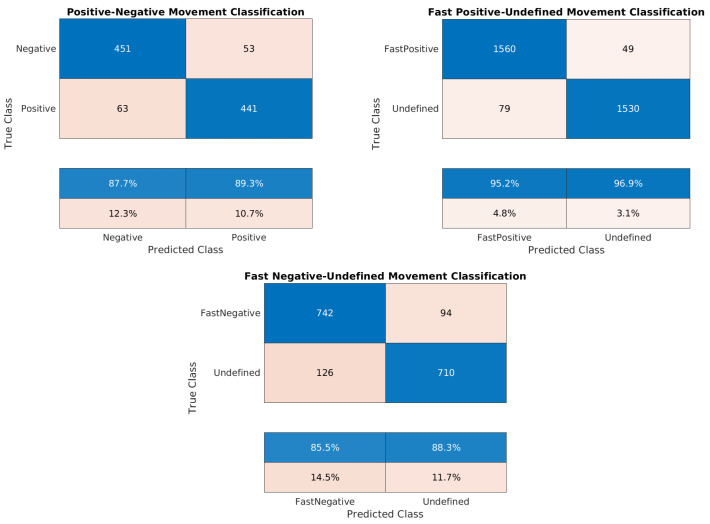
Confusion matrices for the trained models of the Washington dataset: positive/negative movement model, fast positive movement model, and fast negative movement model, respectively.

**Figure 20 sensors-24-02637-f020:**
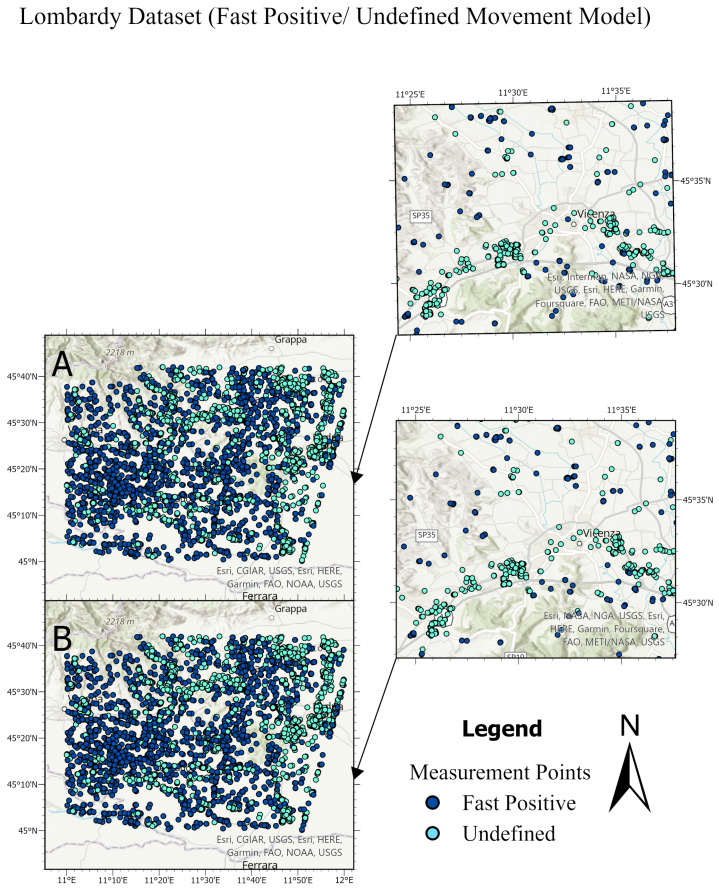
Comparison between the ground truth and the predictions of the test sets for different datasets. Top main figure: Lombardy dataset, showing fast positive and undefined movements. Bottom main figure: Lisbon dataset, displaying positive and negative movements. Each main figure consists of four subfigures. In the **top** subfigure, Subfigure (**A**) presents the ground truth of the Lombardy test set, with Subfigure (**a**) providing a detailed close-up of Subfigure (**A**). Subfigure (**B**) shows the predictions for the Lombardy test set, with Subfigure (**b**) offering a close-up of these predictions. Similarly, in the **bottom** main figure for the Lisbon dataset, Subfigure (**A**) and Subfigure (**a**) focus on the ground truth test set and its close-up, respectively, while Subfigure (**B**) and Subfigure (**b**) depict the predictions and their close-up.

**Figure 21 sensors-24-02637-f021:**
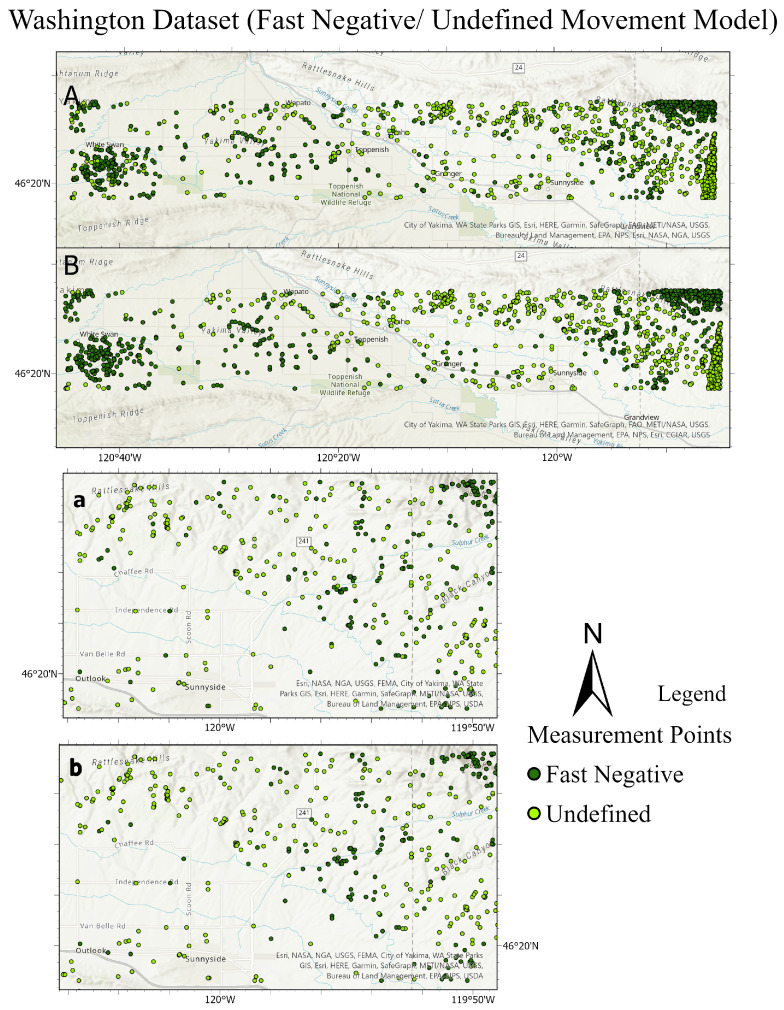
Comparison of the ground truth test set and its predictions for the Washington dataset. The top figure illustrates fast negative and undefined movements, while the bottom figure shows fast positive and undefined movements. In each figure, subfigure **A** depicts the ground truth of the test dataset, and subfigure **a** provides a detailed close-up of **A**. Subfigure **B** presents the predictions for the test dataset, with subfigure **b** offering a close-up view of **B**.

**Figure 22 sensors-24-02637-f022:**
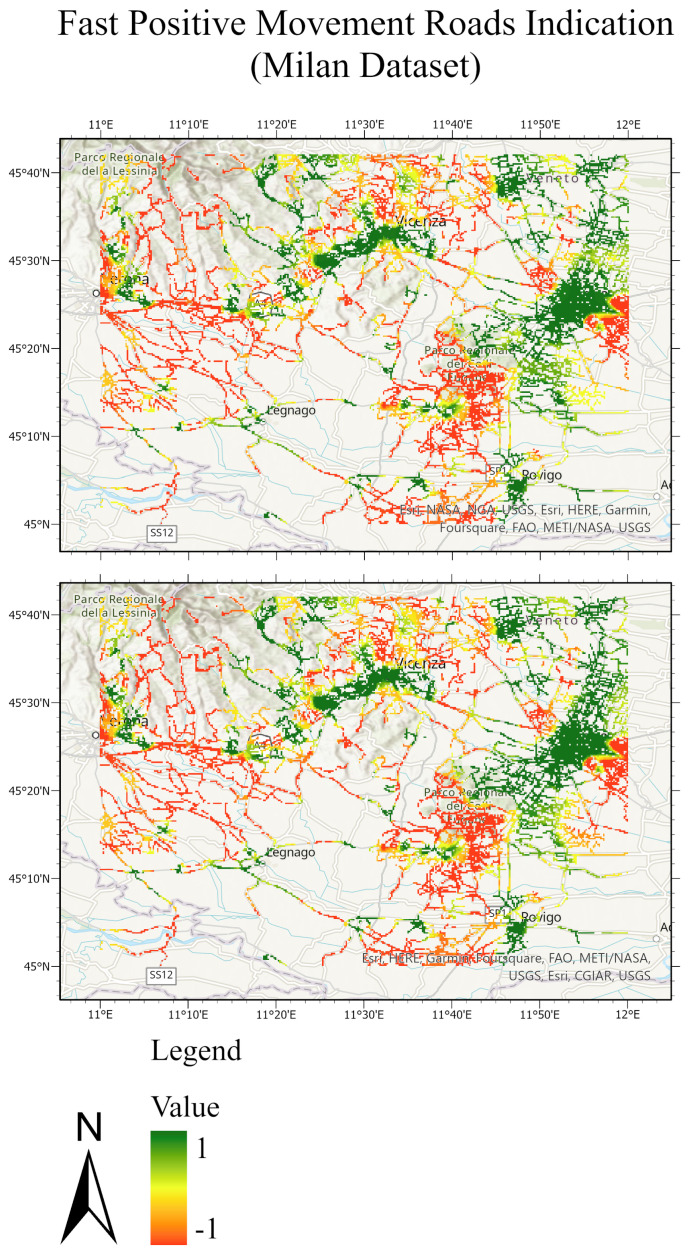
Road network of the Lombardy test dataset. The top figure exhibits the masked roads of the ground truth test set; the bottom figure exhibits the masked roads of the predicted test sets. The value of −1 expresses the fast positive movement while the value of 1 expresses the undefined movement.

**Figure 23 sensors-24-02637-f023:**
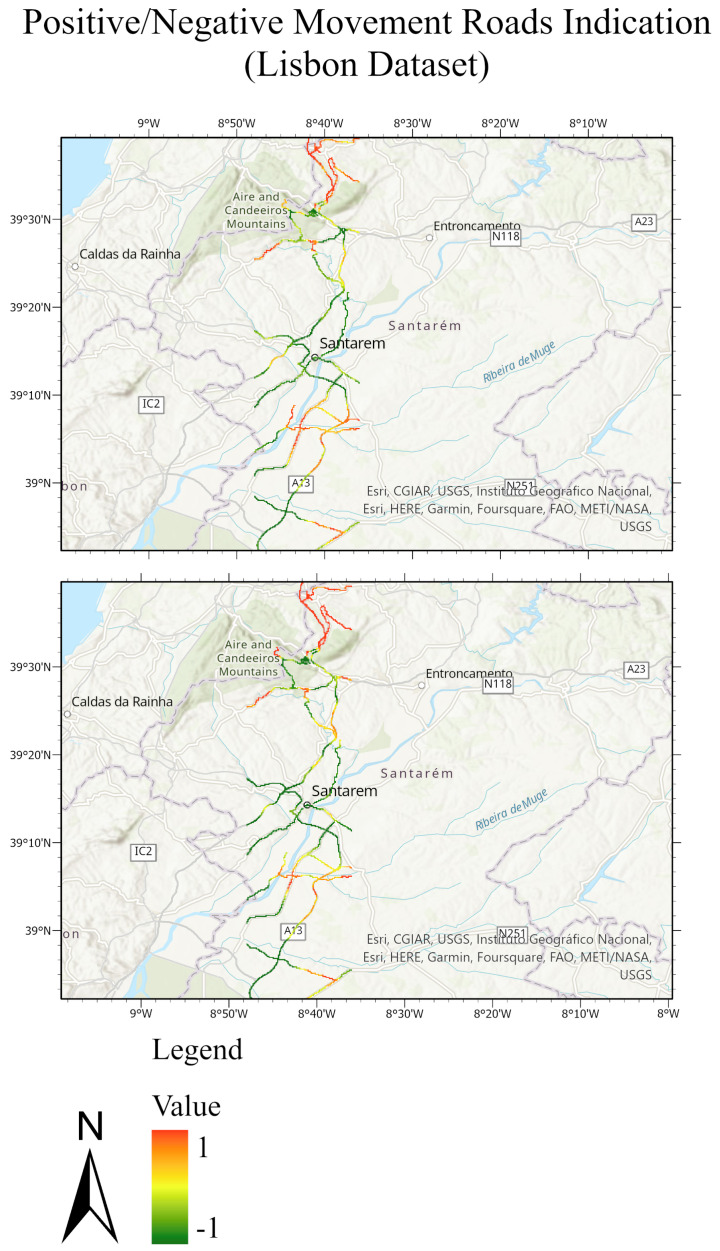
Road network of the Lisbon test dataset. The masked roads of the ground truth test set are shown in the top figure; while the masked roads of the predicted test sets are shown in the bottom figure. The value of 1 expresses the positive movement while the value −1 expresses the negative movement.

**Figure 24 sensors-24-02637-f024:**
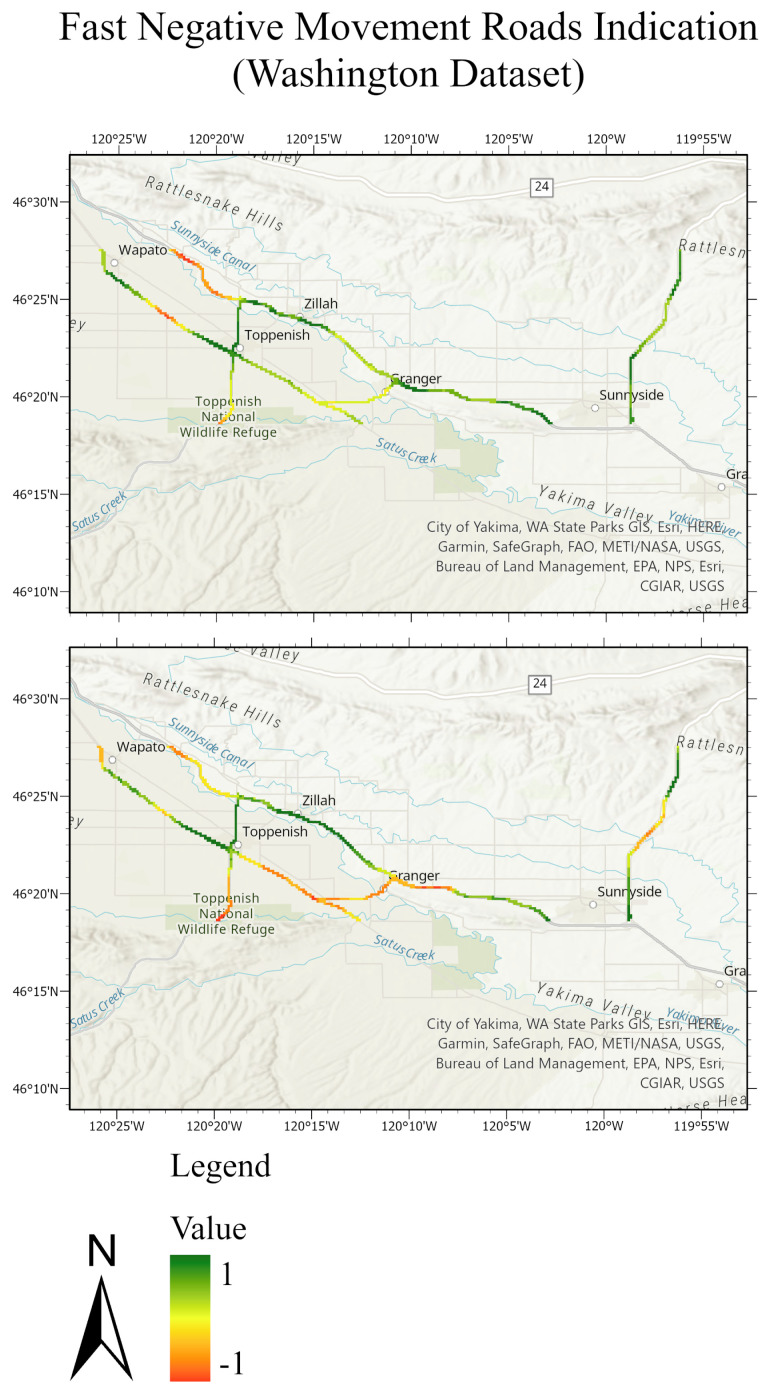
Road network of the Washington test dataset. The top figure represents the masked roads of the ground truth test set; the bottom figure represents the masked roads of the predicted test sets. The value of −1 expresses the fast negative movement while the value of 1 expresses the undefined movement.

**Table 1 sensors-24-02637-t001:** The selected parameters of the P-SBAS analysis for Lombardy (Italy) dataset.

Start Date	7 January 2020
End Date	17 August 2021
Number of Images	50
DEM	SRTM1 arcsec
Temporal Coherence	0.85
Bounding Box	44.943, 8.693
	46.884, 12.231
Orbit Direction	Descending

**Table 2 sensors-24-02637-t002:** The selected parameters of the P-SBAS analysis for the Lisbon (Portugal) dataset.

Start Date	26 January 2018
End Date	27 April 2020
Number of Images	50
DEM	SRTM1 arcsec
Temporal Coherence	0.6
Bounding Box	38.088, −11.124
	39.800, −7.945
Orbit Direction	Ascending

**Table 3 sensors-24-02637-t003:** The selected parameters of the P-SBAS analysis for the Washington dataset.

Start Date	14 October 2016
End Date	28 December 2019
Number of Images	75
DEM	SRTM1 arcsec
Temporal Coherence	0.7
Bounding Box	−121.426, 46.358
	−120.042, 47.167
Orbit Direction	Ascending

**Table 4 sensors-24-02637-t004:** The statistical enumeration of data points obtained from the fast negative/undefined Milan dataset.

Milan Dataset		
**Rate of Movement**	**From 0 To 0.9 Rad**	**From 0.9 To 1 Rad**
FAST	73.64%	26.36%
SLOW	58.80%	41.20%

**Table 5 sensors-24-02637-t005:** The statistical enumeration of data points obtained from the fast negative/undefined Lisbon dataset.

Lisbon Dataset		
**Rate of Movement**	**From 0 To 0.9 Rad**	**From 0.9 To 1 Rad**
FAST	72.81%	27.19%
SLOW	63.68%	36.32%

**Table 6 sensors-24-02637-t006:** The statistical enumeration of data points obtained from the fast negative/undefined Washington dataset.

Washington Dataset		
**Rate of Movement**	**From 0 To 0.9 Rad**	**From 0.9 To 1 Rad**
FAST	37.22%	62.78%
SLOW	26.29%	73.71%

**Table 7 sensors-24-02637-t007:** Positive/negative movement model of the Lombardy dataset.

	Cosine K-NN	1st PS	2nd PS
Number of training samples	10,382	36,596	44,488
Number of testing samples	2596	9148	11,122
Accuracy of validation	75%	93.4%	94.4%
Accuracy of the test set	75.2%	93.3%	94.8%

**Table 8 sensors-24-02637-t008:** Fast negative/undefined movement model of the Lombardy dataset.

	Cosine K-NN	1st PS	2nd PS	3rd PS
Number of training samples	7442	16,878	22,556	25,716
Number of testing samples	1860	4220	5640	6428
Accuracy of validation	85.3%	93.3%	95%	95.6%
Accuracy of the test set	84.4%	93.5%	95%	95.8%

**Table 9 sensors-24-02637-t009:** Fast positive/undefined movement model of the Lombardy dataset.

	Cosine K-NN	1st PS	2nd PS
Number of training samples	952	12,334	29,474
Number of testing samples	238	3084	7368
Accuracy of validation	72.8%	97.8%	98.7%
Accuracy of the test set	70.6%	97.9%	98.8%

**Table 10 sensors-24-02637-t010:** Positive/negative movement model of the Lisbon dataset.

	Cosine K-NN	1st PS	2nd PS	3rd PS	4th PS
Number of training samples	14,596	17,812	20,130	21,576	22,736
Number of testing samples	3650	4454	5032	5394	5684
Accuracy of validation	79.4%	83.4%	85.5%	86.6%	87.2%
Accuracy of the test set	79.4%	82.8%	85.7%	87.1%	87.1%

**Table 11 sensors-24-02637-t011:** Fast negative/undefined movement model of the Lisbon dataset.

	Cosine K-NN	1st PS	2nd PS	3rd PS	4th PS
Number of training samples	16,730	32,978	53,036	72,624	92,320
Number of testing samples	4182	8244	13,260	18,156	23,080
Accuracy of validation	81.3%	91%	94.1%	95.6%	96.4%
Accuracy of the test set	83%	91%	94.2%	95.5%	96.4%

**Table 12 sensors-24-02637-t012:** Fast positive/undefined movement model of the Lisbon dataset.

	Cosine K-NN	1st PS	2nd PS	3rd PS
Number of training samples	2670	9784	29,308	36,618
Number of testing samples	668	2446	7328	9154
Accuracy of validation	83.3%	95.4%	98%	98.2%
Accuracy of the test set	82.3%	95.1%	97.8%	98.4%

**Table 13 sensors-24-02637-t013:** Positive/negative movement model of the Washington dataset.

	Cosine K-NN	1st PS	2nd PS
Number of training samples	3486	3940	4030
Number of testing samples	872	984	1008
Accuracy of validation	86.2%	88.1%	88.4%
Accuracy of the test set	86.1%	88.1%	88.5%

**Table 14 sensors-24-02637-t014:** Fast negative/undefined movement model of the Washington dataset.

	Cosine K-NN	1st PS	2nd PS
Number of training samples	4884	6102	6684
Number of testing samples	1220	1526	1672
Accuracy of validation	81.2%	85.1%	86.1%
Accuracy of the test set	82.3%	84.3%	86.8%

**Table 15 sensors-24-02637-t015:** Fast positive/undefined movement model of the Washington dataset.

	Cosine K-NN	1st PS	2nd PS	3rd PS	4th PS
Number of training samples	2000	3284	7530	11,620	12,876
Number of testing samples	500	823	1882	2906	3218
Accuracy of validation	76%	84.6%	93.6%	95.4%	96.1%
Accuracy of the test set	72.6%	86%	93.3%	95.8%	96%

## Data Availability

Data are contained within the article.

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
