# Peer review of "Learning Ground Displacement Signals Directly from InSAR-Wrapped Interferograms"

_sensors, 2024, doi:10.3390/s24082637_

Round 1

Reviewer 1 Report

Comments and Suggestions for Authors

Synthetic aperture radar data to analyze ground displacements through machine learning, more significantly, the Cosine K-NN model, is examined in this study. Using a pseudo-labeling approach to improve model performance, the research focuses on three different datasets. It offers deformation velocity maps and information about road networks' sensitivity to rapid movements. Artificial intelligence's promise in geospatial analysis is demonstrated by the technique and results, which present a novel approach to extracting useful information from interferograms and coherence maps.

Here are my comments on the improvement of the article:

·       How effectively does the paper address the drawbacks of the conventional SBAS technique and describe the Parallel Small Baseline Subset (P-SBAS) technique?

·       Is using P-SBAS sufficiently justified in terms of efficiency, computational power, and processing speed?

·       In what way does the paper defend the selection of the cloud computing platform Geohazard TEP (G-TEP)? Are the features and advantages of this platform described in detail?

·       How much does the research's scalability and optimization benefit from using cloud computing?

·       To what extent does the article explain why the three datasets (Lombardy, Lisbon, and Washington) were chosen? Are these areas typical of areas where the dangers of geohazards are high?

·       Is there enough justification provided for the settings (start/end date, number of photos, coherence criterion) selected for each dataset?

·       How are the machine learning models trained to classify pixels into Fast Positive, Positive, Fast Negative, Negative, or Undefined movements?

·       The following studies are recommended to be appropriately cited:

[1]  https://doi.org/10.1016/j.rsase.2022.100905

[2] https://doi.org/10.1007/s12524-023-01707-y

[3] https://doi.org/10.1109/ICARES60489.2023.10329911

·       Are the thresholds for movement rate classification (0 cm/year, ±0.7 cm/year) justified, and how sensitive are the models to these thresholds?

·       According to the testing accuracy, how reliable are the machine learning models, and how well do they generalize to nearby regions?

·       How does using pseudo-labeling to improve model performance affect the outcomes, and is it sufficiently explained in the article?

·       How well does the article justify the choice of the Cosine k-nearest Neighbor model as the best-performing model, and what are the implications of this choice?

·       What challenges or limitations does the study encounter in achieving accuracy, and how might these impact the reliability of the results?

Author Response

Dear Reviewer,

Thank you for your time and effort. We have carefully considered each comment and suggestion and have made substantial revisions to the manuscript. We believe these changes address the concerns raised and significantly enhance the quality and clarity of our work.

We have attached a detailed response to your comments, which includes point-by-point responses and indicates where changes were made in the manuscript.

We hope that our revisions adequately address the concerns you raised and thank you for considering our revised manuscript for publication in the esteemed MPDI Journal for Special Issues.

Best Regards 

Reviewer 2 Report

Comments and Suggestions for Authors

The following are a few obvious issues in current version.

The journal is on sensor. The paper is titled “Learning displacement signals directly from InSAR wrapped

interferograms using Sentinel-1 and artificial intelligence”. Explain how the main contribution of the paper is important for sensors.

“InSAR” in the title needs explanation before using it.

“Interferogram” in the title needs explanation before using it.

“Sentinel-1” in the title needs explanation before using it.

“artificial intelligence” in the title needs to be more specific, relevant to own innovation.

The main research has been addressed by Table 1 in page 3, Table 2 in page 5, Table 3 in page 6, Fig 6 in page 8, Fig 8 in page 12, Fig 9 in page 12, Table 7, 8 and 9  in page 19, Table 10-14  in page 20,

Fig 17-18 in page 21, Fig 19 in page 22, et al.

Please make sure the references are in the correct format.

The conclusions seem consistent with the main text presented.

Lack 2024, 2023 references.  Please discuss more relevant recent work.

The paper cited 32 papers, please explain how each of them directly relevant and necessary for this paper.

Please explain originality and advantages of own solution more accurately. Please explain explicitly the original innovation in this paper and advantages of own innovation comparing with existing relevant solutions.

Explain in good technical detail the computational complexity of the solutions.

Explain the validity and generalizability of experimental results in good technical details.

Please explain with more technical details on limitations of this work and future work.

Please proofread the whole paper. Please use concise accurate terms instead of general vague terms.

Keyword P-SBAS needs to be explained at first appearance.

Keyword Machine Learning is too general. Please use concise accurate terms instead of general vague terms.

Keywords need to be more specific, relevant to own innovation.

Equation on page 14 is existing.

The paper lacks computing details.

Please explain how own computing is original and better than existing solutions.

Table 1 in page 3 , Table 2 in page 5, Table 3 in page 6   presented the parameter settings. Please explain the choice of parameter settings.

For Table 7, 8 and 9  in page 19, Table 10-15 in page 20,  please explain value ranges, such as whether all values are percentages, whether all values are of same scale, and whether the values are normalized or standardized.

Comments on the Quality of English Language

The writing can be improved

Author Response

(The authors gave the same response as above.)

Round 2

Reviewer 1 Report

Comments and Suggestions for Authors

The authors have addressed all the comments. It may be accepted for publication. A new title, "Learning ground displacement signals directly from InSAR wrapped interferogram," may be accepted as requested by the authors.

Author Response

Dear Reviewer,

We express our sincere gratitude to you for your insightful feedback, which has significantly contributed to enhancing the quality of our paper.

Best Regards

Reviewer 2 Report

Comments and Suggestions for Authors

The new version is improved, but did not fully address previous version review reports.

Please explain with more technical details on sensors used and computing innovations of sensor data processing. 

Please explain with more technical details on selection of computing solutions. 

Please explain with more technical details on selection of parameters and initial value assignment of selected parameters. 

Comments on the Quality of English Language

Proofread the file and make sure the used terms are accurate.

Author Response

Dear Reviewer,

We express our sincere gratitude to you for your insightful feedback, which has significantly contributed to enhancing the quality of our paper. We have replied to the comments in the attached file and hope that now we have addressed them fully.

Best Regards
